# Unsupervised learning of categories

## Abstract

Humans are known to reason using logic and abstract categories, and yet most state of the art neural models use continuous distributed representations. These representations offer impressive gradient-based learning capabilities, but it is often difficult to know what symbolic algorithm the network might implicitly be implementing, if any. We find that there are representational geometries that naturally suggest a symbolic structure, which can be expressed in terms of binary components. We show that we can recover this structure by fitting the geometry of this binary embedding to the representational geometry of the original objects. After establishing general facts and providing some intuitions, we present two algorithms that work on low-rank or full-rank data, respectively. We assess their reliability on simulated data, and then use them to interpret neural word embeddings, in which we expect a compositional structure.

## 1 Introduction

In biological and artificial learning systems, compositional structure is important to flexible behavior, yet it is difficult to detect at the representational level. Neural representations are rarely factorized into purely-selective concept neurons; when there is a neat conceptual structure it is most often embedded into high-dimensional neural modes (Kaufman et al., 2022; Higgins et al., 2021; She et al., 2021; Bernardi et al., 2020; Courellis et al., 2024). Modern machine learning systems also use distributed continuous representations, which are rarely factorized even when symbolic or compositional structure are explicitly incorporated into the model (Altabaa et al., 2024; Rigotti et al., 2022; Mao et al., 2019).

Identifying latent structure can enable compact descriptions of neural computation. This is true for biological systems (Bernardi et al., 2020; Courellis et al., 2024) and artificial. Huben et al. (2024) were recently able to use sparse autoencoders to find causally-relevant factors in language model representations. While the factors learned by such approaches are continuous they are often analyzed discretely (is it active or not), which can introduce "quantization errors" to the reconstruction and may be misleading, in that they will not accurately reflect the true latent structure. For that reason, we see advantage in having factors which are categorical by construction.

In addition to interpreting fixed models, there might be some value to logical representations when building a model. An early critique of connectionism was its inability to account for systematic generalization (Fodor and Pylyshyn, 1988). Despite remarkable and unanticipated advances, modern connectionist systems still struggle with abstract reasoning and out-of-distribution generalization (Chollet, 2019; Mitchell et al., 2023; Moskvichev et al., 2023; Anil et al., 2022). Being able to efficiently convert learned continuous representations into a symbolic equivalent can help leverage advantages of both gradient-based and symbolic computation (Mao et al., 2019; Koh et al., 2020).

We will show that the discovery of latent categories can be formulated as a binary matrix factorization. Given a representation, $\mathbf{X}$, we will try to find a binary representation, $\mathbf{S}$, which encodes an assignment of items to logical variables in such a way that preserves distances. We refer to these logical variables as 'concepts'. Many structures–including analogies, clustering, hierarchy, ordering, and hybrids of these–can be captured by binary concepts. However, binary matrix factorization is a difficult combinatorial problem, so, in addition to introducing it as a tool for concept discovery, we offer a new efficient algorithm.

**Prior work**  We are in many ways motivated by ongoing interest in symbolic capabilities of continuous representations. There is a long tradition of neuro-symbolic paradigms which do this explicitly; for example, tensor product representations (Smolensky et al., 2022), and more recently transformer-inspired architectures (Altabaa et al., 2024; Mao et al., 2019). Yet compositional and symbol-like vectors can appear in more standard connectionist architectures (Zhou et al., 2024), and also in biology, where it is hidden by lack of mechanistic understanding and requires bespoke analyses to discover. In such a case it could be very useful to have a systematic way of making explicit underlying compositional structure.

The field of mechanistic interpretability offers many ideas and methods related to continuous representations of categorical structure. Our formalism and method can be seen as operationalising the linear representation hypothesis (Park et al., 2024; 2023), by "unembedding" a linear representation of categorical variables. In particular our use of orthogonal weights fits nicely in that framework. A similar point of view has been taken by recent work on the representation of sparse variables in language models (Elhage et al., 2022; Huben et al., 2024), and we aim to enable a similar discovery process for categorical variables.

In computer science, what we seek has been called locality-sensitive hashing (Andoni and Indyk, 2006). Salakhutdinov and Hinton (2009) proposed a binary latent variable model for similarity-preserving hashing of documents, and Mena and Ñanculef (2019) tackled the same problem with a variational autoencoder. At the level of the generative model these approaches are very similar to ours, but with different algorithms and goals, as these methods are often non-linear and do not always seek interpretable features.

In the community detection and applied math literature, our specific factorization problem has been studied as (semi) binary matrix factorization (SBMF) or binary component decomposition (BCD). Remarkably, in special cases an algebraic solution is available via tensor decomposition (Sørensen et al., 2021), but it is highly sensitive to violations of its assumptions. There are several optimization-based approaches (Zhang et al., 2007; Kolomvakis and Gillis, 2023; Sørensen et al., 2022) which are generally built around the assumption of very low-rank data, and thus may not be applicable in the general case. Our specific model formulation closely follows that of Kueng and Tropp (2021), and we substantially extend the scope of their model by fitting more general structures to to noisy data.

**Contributions**  We extend previous work on this topic both conceptually and practically. While others have formalised the representation of categorical structure in various ways, they often focus on specific kinds of structure and developed appropriately sophisticated formalisms. Framing the search for compositional structure as a simple matrix factorization makes the general problem conceptually tractable, and can be used as a starting point for more structured analyses. On the technical side, the literature on SBMF has restricted itself to the case of identifiable factorizations, and we substantially extend the scope of that work by applying mild regularization, and provide an overlooked connection to graph structure as a visualisation tool. By bringing insights from the literature on SBMF to the field of interpretability, we can offer a new perspective on an important and under-studied problem.

Practically speaking, we try to marry the advantages of the matrix factorization and autoencoder approaches, by providing a general heuristic algorithm without restrictive requirements on the data, which nevertheless can take advantage of the structure of the problem for substantial gains in efficiency and ground truth recovery. Due to the straightforward implementation, our algorithms can be more easily integrated into differentiable systems as well. For instance, "concept bottleneck models" (Koh et al., 2020), in which the eponymous concepts are normally hand-labeled. In general, we hope that the simplicity and effectiveness of our approach can inspire deeper exploration of this problem.

## 2 Problem formulation

Our factorization approach tries to identify any compositional structure that is present in the representational geometry. For example, the transformation from a white square to a shaded square is similar to the transformation from a white circle to a shaded circle. For some representational geometries, this form of compositionality is reflected by vector

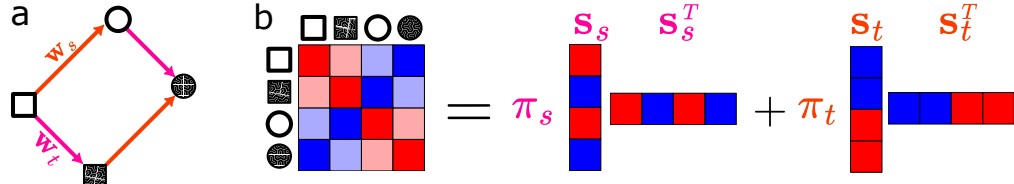

Figure 1: Illustration of the matrix factorization. (**a**) A representation with compositional structure in which shared feature vectors are added together. (**b**) The kernel matrix, which we model as a conic (i.e. non-negative linear) combination of rank-one binary matrices.

additivity. For example, in Fig. 1a we can get from the white square to the shaded square by adding the $\mathbf{w}_t$ "feature" vector associated with $t$. This is the same vector that allow us to move from the representation of the white circle to the representation of the black circle. Thanks to the additivity structure of the geometry, we can also predict what we will see when we add both the $\mathbf{w}_s$ and the $\mathbf{w}_t$ vectors. This kind of conceptual navigation is what we will try to discover from the data.

Our criterion will be geometry preservation. If the representation of the data is composed of some shared concepts, then the similarity of a pair of items should be related to the number of categories they both belong to. More precisely, we will model the dot product between each pair of observations by the positive weighted sum of the number of shared concepts:

$$\mathbf{X}\mathbf{X}^T \sim \sum_{\alpha=1}^{b} \pi_\alpha \mathbf{s}_\alpha \mathbf{s}_\alpha^T \tag{1}$$

where the rows of $\mathbf{X} \in \mathbb{R}^{p \times n}$ are the representation of the data, $\mathbf{s}_\alpha \in \{0,1\}^p$ is the binary assignment of each item to the concept $\alpha$, and $\pi_\alpha \in \mathbb{R}_+$ is the weight the concept has in the representation (Fig.1b). We will refer to the matrix whose columns are the concept vectors by $\mathbf{S}$, the binary embedding of $\mathbf{X}$. By "$\sim$"ing we mean distance preservation, which will be made precise in the following section.

A distance-preserving embedding can be seen as implicitly inverting a linear encoding model. In matrix terms, we model the data as the product of a real and a $\{0,1\}$-valued matrix:

$$\mathbf{X} \sim \mathbf{W}\mathbf{S}^T$$

where the columns of $\mathbf{W} \in \mathbb{R}^{n \times b}$ are the feature vectors associated with each concept. If the columns of $\mathbf{W}$ are orthogonal, then the $\pi$ values above will correspond to their squared norms. Rather than fitting $\mathbf{W}$ jointly with $\mathbf{S}$, which increases the number of parameters and can be very sensitive to initialization (Sørensen et al., 2022), we fit $\mathbf{W}$ after the fact using either Procrustes or ordinary least squares regression.

**Existence and uniqueness**   Without an orthogonality constraint, clearly all data, $\mathbf{X}$, can be decomposed with an exact binary embedding; just set $\mathbf{W} = \mathbf{X}$ and $\mathbf{S} = \mathbb{I}$. With orthogonal $\mathbf{W}$, not all $\mathbf{X}$ can be factorized exactly, and it is NP-hard to check for a particular $\mathbf{X}$ (Deza and Laurent, 1997). Nevertheless, most data is 'fairly close' to an exactly-embeddable representation (Laurent and Poljak, 1996), and we give some small examples in Figure 2. Among our examples, the square, tetrahedron, and tree are exactly embeddable, while the grid and hexagon are best approximations (as defined in the following section).

Exact or approximate, the optima are rarely unique[1]. A necessary and sufficient condition for uniqueness would be NP-hard to check (Deza and Laurent, 1997), but several sufficient conditions have been derived of varying restrictiveness and complexity (Kueng and Tropp, 2021; Sørensen et al., 2022). A crude intuition: the higher-dimensional the data, the more possible solutions. Two extreme examples are the $b$-cube and the $p$-simplex. A hypercube has $p$ points in $b$ dimensions, and its binary representation is unique; meanwhile, a $p$-simplex (every point equidistant) has $p-1$ dimensions, and a tremendous number of possible

---

[1]Unique up to flipping all bits of a concept, which is a trivial symmetry of our model.

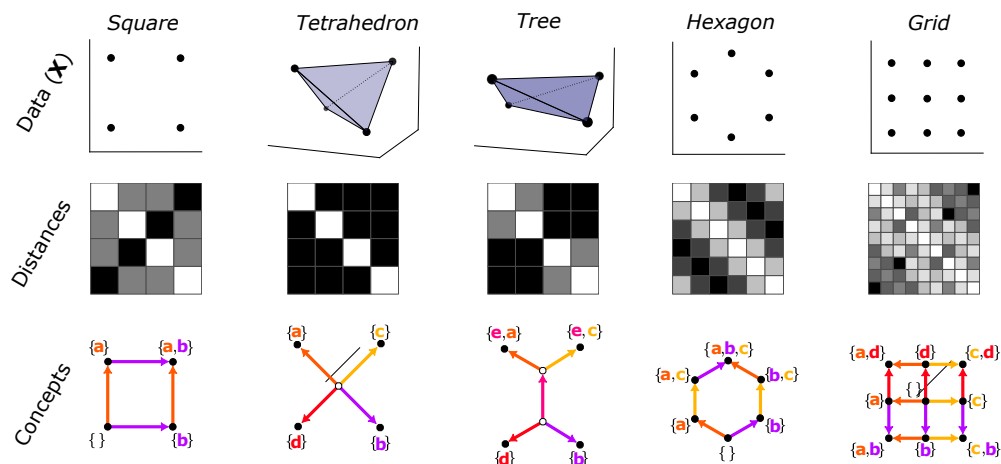

Figure 2: Examples of some categorical structures recoverable from geometry. The concepts are found by optimizing Equation 3 with brute force (i.e. setting $\mathbf{S} = \{0,1\}^p$) and sparsity regularization. The graphs (see Section 2) are drawn by manual inspection. Notice that the concept labels (in curly brackets) can be gotten by cutting the graph at the corresponding edges, and labelling the partition on the arrow side. We chose the source nodes arbitrarily.

representations, including the identity matrix and all Hadamard matrices. Because of this, we argue for a sparsity inductive bias in Section 2.1.

**Graphical representation** High-dimensional binary vectors can be hard to interpret, and we would like a visualization tool. Hierarchical clustering can be visualized with a dendrogram, *i.e.* a tree on which observations are leaves and cluster assignments can be recovered by cutting the tree at a certain depth. Just as we generalize hierarchical clustering, we can generalize the dendrogram.

Our goal will be to draw a graph in which (1) distances match those in $\mathbf{S}$ and (2) the concepts themselves can easily be recovered. Given a set of $b$ concepts, $\mathbf{S}$, we can isometrically embed each point (row of $\mathbf{S}$) on a 'partial cube', i.e. an isometric subgraph of the $b$-bit hypercube. In this representation (e.g. Fig.2, bottom row), nodes are connected by an edge when they differ by one and only one concept (i.e. only one component of the concept vector differs). Some nodes of the hypercube which do not correspond to observations might be necessary to form the graph. Concepts can be read out by cutting (i.e. removing) all edges corresponding to the same concept, and for visualization it is often easier to color the edges to know which to cut (as we have done in Fig.2). By analogy to a dendrogram (which is itself a type of partial cube) we will call this kind of graph an 'analogram'.

Every partial cube has an easily-obtained unique binary representation (Deza and Laurent (1997), and see the Appendix A.3 for explicit construction), but there is a combinatorial number of graphs whose binary representation is $\mathbf{S}$. The analogram should be the smallest such graph. Unfortunately, as often the case in this problem, finding such an analogram is NP-hard and it needn't even be unique (Knauer and Nisse, 2019). Nevertheless, we made a heuristic algorithm that works well on moderately-sized $\mathbf{S}$, and always on trees (see A.3).

## 2.1 OBJECTIVE

Since an exact fit is not always possible (or desirable, in the presence of noise), we must define goodness of approximation. The centered kernel alignment (CKA) is commonly used to measure representational similarity in machine learning and neuroscience (Kornblith et al., 2019), and intuitively measures the similarity of the two representational geometries. It is related to the decodability of one representation from another, but it is translation and

scale invariant, which is desirable in a measure of geometric similarity (but see Davari et al. (2022); Harvey et al. (2023)).

The CKA is the cosine similarity of the flattened, double-centered Gram matrices. Specifically, if we have $p \times p$ Gram matrices $\mathbf{K} = \mathbf{X}\mathbf{X}^T$, and $\mathbf{Q} = \mathbf{S}\mathbf{\Pi}\mathbf{S}^T$, then double-centering means computing the dot products after subtracting the mean from each dimension. If we define the centering matrix $\mathbf{H} = \mathbb{I}_p - \frac{1}{p}\mathbf{1}_p$, then we will denote the centered features and kernels by $\bar{\mathbf{X}} \doteq \mathbf{H}\mathbf{X}$ and $\bar{\mathbf{K}} \doteq \mathbf{H}\mathbf{K}\mathbf{H}$, respectively.

So, the problem we want to solve is

$$\underset{\mathbf{Q}=\mathbf{S}\mathbf{\Pi}\mathbf{S}^T}{\arg\max} \quad \frac{\mathrm{tr}(\bar{\mathbf{K}}\bar{\mathbf{Q}})}{\sqrt{\mathrm{tr}(\bar{\mathbf{K}}\bar{\mathbf{K}})\,\mathrm{tr}(\bar{\mathbf{Q}}\bar{\mathbf{Q}})}} \tag{2}$$

For practical reasons, it will be easier to minimize a distance function than to maximize the cosine. While it is not generically the case for constrained optimization, since our constraining set is closed under positive scaling this is equivalent to minimizing the mean squared error between the centered kernel matrices:

$$\underset{\mathbf{S}\in\{0,1\}^{p\times b},\boldsymbol{\pi}\in\mathbb{R}_+^b}{\arg\min} \quad \|\bar{\mathbf{S}}\mathbf{\Pi}\bar{\mathbf{S}}^T - \bar{\mathbf{X}}\bar{\mathbf{X}}^T\|_F^2 \tag{3}$$

which can be solved directly, when $p$ is small enough, by fixing $\mathbf{S}$ to include all binary vectors in $\{0,1\}^p$ and optimizing over $\boldsymbol{\pi}$. However when $p$ is greater than $\sim 10$ this is very impractical, and so heuristic algorithms are needed.

### 2.1.1 Regularization

There could be an astronomical number of optima for a given geometry, so we need an inductive bias to break the tie. We would like to select for the simplest structure that fits the data. Using the minimal number of concepts might seem like a reasonable choice, but this will favor dense concepts which can lead to more complex relationships between them (A.2). Instead we should try to minimize the size of the analogram (Section 2). Since that is very hard to do directly, we notice that the simpler graphs tend to have sparser concepts, and propose regularizing for sparsity.

To encourage sparsity, we add linear (in $\mathbf{S}$) term to the loss: $\mathbf{1}^T\mathbf{S}\mathbf{1}$. This does not guarantee identifiability[2], but, intuitively, there are fewer sparse concepts than dense ones, and so we try to reduce the space of solutions by asking for the sparsest concepts. Sparsity is a common requirement for solving otherwise ill-posed inverse problems (Donoho, 2006) and, as an inductive bias, has also been found to parsimoniously explain structure learning in humans (Lake et al., 2018; Kemp and Tenenbaum, 2008).

## 3 Optimization

Being a challenging combinatorial problem, we cannot expect efficient solutions that work in every situation. There are already remarkably effective approaches for very low-rank data, but they do not always fail gracefully when their assumptions are violated. Here, we develop an algorithm which can exploit low-rank structure when it exists, and another to handle the general case. Note that both algorithms can in principle work on any positive semidefinite matrix, and are thus quite general.

### 3.1 Rejection sampling for low-rank data

When the dimensionality of the data is sufficiently small, Kueng and Tropp (2021) and Kolomvakis and Gillis (2023) provide efficient algorithms based on randomly sampling columns of $\mathbf{S}$. These methods are based on the following observation: If the data admits a factorization of the form $\bar{\mathbf{X}} = \bar{\mathbf{S}}\mathbf{W}^T$, and $\mathbf{W} \in \mathbb{R}^{n \times b}$ has full column rank, then,

---

[2]We provide detailed examples in A.2 of the kinds of degeneracy that can exist. For the curious: in our illustrations, the sparsest solution is unique, but this is not generally true. A counter example is the 16-vertex cross polytope geometry, which has two solutions of equal sparsity.

since the linear mapping can be inverted, each column of $\bar{\mathbf{S}}$ is also in the image of $\bar{\mathbf{X}}$. So, these algorithms randomly search for such vectors $\mathbf{s} \in \mathrm{im}(\bar{\mathbf{X}}) \cap \{0, 1\}^p$. When the rank, $r$, is very small, it is possible to use exhaustive search since there are at most $2^r$ such binary vectors (Slawski et al., 2013). In general, more efficient methods are required.

The aforementioned algorithms are able to search efficiently by assuming that the $\mathbf{s}$ vectors are 'Schur-independent'. This means that the set of bitwise exclusive-or vectors of all pairs of concept vectors, $\{\mathbf{1}\} \cup \{\mathbf{s}_\alpha \oplus \mathbf{s}_\beta\}_{\alpha,\beta=1}^b$, is linearly independent. In that case, because of the special structure of the set of correlation matrices (Laurent and Poljak, 1996; Kueng and Tropp, 2021), each $\mathbf{s}$ can be found via semidefinite programming (SDP). But in the general case, when the rank is still low but not low enough, the SDP is not guaranteed to have a rank-one optimum, and would require a heuristic rounding step. In such a situation it might not be worth the computational burden of solving an SDP, which has $O(p^2)$ variables, and so we propose a method based on Hopfield networks.

In the presence of noise, the true concept vectors might not be exactly in the image of $\bar{\mathbf{X}}$, but just the closest among nearby binary vectors. If $\mathbf{U}$ are the right singular vectors (with non-zero singular values) of $\mathbf{X}$, then we expect the true concept vectors, $\mathbf{s}$, to be local maxima of $E(\mathbf{s}) = \mathbf{s}^T \mathbf{U}\mathbf{U}^T\mathbf{s}$. This is precisely (the negative of) the energy function of a Hopfield network (Hopfield, 1982), which we can maximize by iteratively updating $\mathbf{s}_t \leftarrow \mathrm{sign}(\mathbf{U}\mathbf{U}^T\mathbf{s}_{t-1})$ from some initial guess.

There is an interesting connectionist interpretation of this procedure. Let us imagine the concept $\mathbf{s}$ is the binary response pattern of neuron to the whitened inputs – then the Hopfield updates amount to Hebbian plasticity. If the weights of the neuron are $\mathbf{m}$, then the update above tells us that it should be set to $\mathbf{m}_i = \sum_j \mathbf{U}_{i,j}\mathbf{s}_j$, which is a Hebbian rule. Taking this interpretation, we call this algorithm the binary autoencoder (BAE). It works by randomly sampling $\mathbf{s}$ vectors according to Algorithm 1, with a tolerance parameter $\epsilon$. To improve efficiency, we can also try to discourage any accepted $\mathbf{s}$ from being resampled by subtracting some part of its projection onto $\mathbf{U}$: i.e. applying $\mathbf{U} \leftarrow (\mathbb{I} - \frac{\lambda}{p}\mathbf{s}\mathbf{s}^T)\mathbf{U}$ for some $\lambda \geq 0$.

**Numerical validation**   We find that this simple algorithm performs very well on simulated low-rank data. For $p$ points, we draw $\sqrt{2p}$ random $\mathbf{s}$ vectors, which almost certainly admit a unique decomposition, and embed them in $d$-dimensions with a random orthogonal matrix. We then add iid Gaussian noise to achieve a specified signal-noise ratio (SNR).

We compare against two baselines: a two-layer tanh autoencoder, and a variational autoencoder with Bernoulli latents (Mena and Ñanculef, 2019). The two differ primarily in the a binary entropy term present in the loss of the VAE; we also include sparsity regularization in both. We do not compare against algorithms for SBMF since they do not apply to the data we will test in the next two sections, and have also recently been benchmarked on the same type of simulations we are running here (Kolomvakis and Gillis, 2023). Compared to the two gradient-based baselines, the sampling algorithm is better at recovering ground truth across the full range of sizes, and about an order of magnitude faster (Fig. 3a). We note that the VAE has slightly better ground truth recovery than the tanh autoencoder, while doing substantially worse at reconstructing the overall geometry, indicating only partial recovery.

| **Algorithm 1** Rejection sampler (BAE) | **Algorithm 2** Hopfield coordinate descent |
|---|---|
| 1: **function** SAMPLE($\mathbf{U} \in \mathbb{R}^{p \times r}$, $\epsilon > 0$) | 1: **function** HOP($\mathbf{S}$, $\mathbf{X} \in \mathbb{R}^{p \times n}$, $\mathbf{x} \in \mathbb{R}^n$, $T > 0$) |
| 2: $\quad$ $\mathbf{s} \sim p_0(\mathbf{s} \in \{0, 1\}^p)$ | 2: $\quad$ $\mathbf{J} = $ Eq. 9 |
| 3: $\quad$ **while** not converged **do** | 3: $\quad$ $\mathbf{h} = $ Eq. 10 |
| 4: $\quad\quad$ $\mathbf{s} \leftarrow \Theta[\mathbf{U}\mathbf{U}^T\mathbf{s}]$ | 4: $\quad$ $\mathbf{s} \sim p_0(\mathbf{s} \in \{0, 1\}^b)$ $\quad\quad$ ▷ *Initialise* $\mathbf{s}$ |
| 5: $\quad$ **if** $\|\mathbf{U}^T\mathbf{s}_t\|_2^2 \geq (1 - \epsilon)p$ **then** | 5: $\quad$ **for** $i = 1, ..., b$ **do** |
| 6: $\quad\quad$ **Return** $\mathbf{s}$ | 6: $\quad\quad$ $c = \mathbf{h}_i + (\mathbf{s}_i - \frac{1}{2})\mathbf{J}_{ii} - \sum_{j \neq i}\mathbf{J}_{ij}\mathbf{s}_j$ |
| 7: $\quad$ **else** | 7: $\quad\quad$ $\mathbf{s}_i \sim$ Bernoulli($\eta = c/T$) |
| 8: $\quad\quad$ **Return** SAMPLE($\mathbf{U}$, $\epsilon$) | 8: $\quad$ **return** $\mathbf{s}$ |

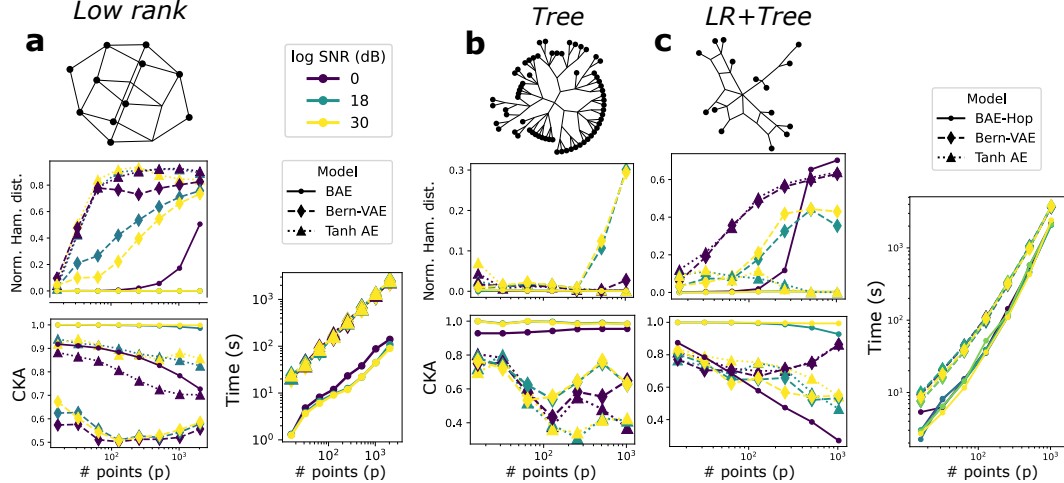

Figure 3: Numerical experiments. All plots are the average of 12 random seeds, and optimized over hyper-parameters. (**a**) Data generated by Schur-independent concepts. We report the average Hamming distance of the discovered concepts, $\mathbf{S}$, to the nearest ground truth, $\mathbf{S}^*$, minimized over permutations of the columns of $\mathbf{S}$, and normalized by the size of the data. (**b**) Data generated by hierarchically-structured concepts. (**c**) Same as before, but with data generated by a mixture of Schur-independent and hierarchical concepts.

### 3.2 ITERATIVE REFINEMENT FOR FULL-RANK DATA

When $\mathbf{X}$ is full rank, every concept is in $\text{im}(\bar{\mathbf{X}})$, and so we cannot use Algorithm 1. We must instead optimize our objective function (Eq. 3) over $\{0,1\}^{p \times b} \times \mathbb{R}_+^b$, which is hard. While we leave the full derivation to the Appendix (A.4), we show that optimizing one row of $\mathbf{S}$ at a time with coordinate descent makes it a more tractable problem.

To first establish notation: Assume that we are updating the $p^{\text{th}}$ row. We denote the first $p-1$ rows by $\mathbf{S} \in \{0,1\}^{p-1 \times b}$, and the row being updated by $\mathbf{s} \in \{0,1\}^b$. The corresponding rows of the data are $\mathbf{X}$ and $\mathbf{x}$, respectively. Denote the row-wise mean of $\mathbf{S}$ by $\langle \mathbf{s} \rangle \doteq \frac{1}{p-1}\mathbf{S}^T \mathbf{1}$.

To minimize the loss (3) with respect to $\mathbf{s}$, we can minimize the following function:

$$\mathcal{L}(\mathbf{s}) = \mathbf{s}^T \mathbf{J}\mathbf{s} - 2\mathbf{h}^T \mathbf{s}$$

$$\mathbf{J} = 2\bar{\mathbf{S}}^T \bar{\mathbf{S}} + t\langle \tilde{\mathbf{s}} \rangle \langle \tilde{\mathbf{s}} \rangle^T \tag{4}$$

$$\mathbf{h} = \mathbf{J}\langle \mathbf{s} \rangle + t\langle 1 - \mathbf{s} \rangle^T \langle \mathbf{s} \rangle \langle \tilde{\mathbf{s}} \rangle - t\bar{\mathbf{x}}^T \bar{\mathbf{x}} \langle \tilde{\mathbf{s}} \rangle + 2\mathbf{S}^T \bar{\mathbf{X}}\bar{\mathbf{x}} \tag{5}$$

where $\langle \tilde{\mathbf{s}} \rangle \doteq 2\langle \mathbf{s} \rangle - 1$ and $t \doteq \frac{p-1}{p}$. The above equations exclude $\mathbf{\Pi}$ for legibility, but the term can be added back in as shown in the full derivation (A.4). We note that a convex relaxation is possible, which we describe in (A.5), but in our experience it is less robust.

Because $\mathcal{L}$ is a quadratic function of $\mathbf{s}$, there are many heuristic tools available. Punnen (2022) provides a survey of exactly-solvable cases and approximate algorithms, on top of which we mention recent work on iterative submodular approximation (Konar and Sidiropoulos, 2019) and old work on Hopfield networks (Hopfield and Tank, 1985). The Hopfield approach is intriguing, since the weights and offsets are formed of Hebbian terms ($\bar{\mathbf{S}}^T \bar{\mathbf{S}}$ and $\mathbf{S}^T \mathbf{X}$) and rank-one terms ($\langle \tilde{\mathbf{s}} \rangle \langle \tilde{\mathbf{s}} \rangle^T$) which could be implemented with local learning rules in a neural or neuromorphic system[3]. Furthermore, Hopfield updates just use repeated matrix-vector multiplication, so we can profit off the potentially extreme sparsity of $\mathbf{S}$, given our chosen regularization. For these reasons it is our method of choice (Algorithm 2).

The Hopfield-style coordinate descent can very naturally be combined with the rejection sampling algorithm in order to produce a general-purpose algorithm for low-but-not-full

---

[3]The middle terms of $\mathbf{h}$ are not so clearly local, and may need auxilliary neurons to implement.

rank data. $\mathbf{S}$ can be initialized by sampling $b$ concepts from Algorithm 1, and then we can iterate between steps of Algorithm 2 and projections onto the image of $\bar{\mathbf{X}}$.

**Numerical validation**   We test this algorithm on hierarchical concepts, as well as a hybrid low-rank and hierarchical structure. To generate this data, we recursively partition the points by randomly choosing a number of partitions between 2 and 4, defining a concept for each partition, and then repeating within each partition. The same noise model was used as before. The hybrid structure is just the concatenation of randomly-sampled concepts and the hierarchical ones. We use the iterative refinement algorithm with an exponential annealing schedule ($T = \gamma^t T_0$, with $\gamma = 0.8$ and $T_0 = 5$) run for 100 iterations. We note that the run time of the algorithm is depends on the sparsity of the discovered solution, since we use sparse matrices, but is roughly the same for both kinds of data.

In comparison with gradient-based autoencoders, the Hopfield approach does better at recovery and geometry matching at all noise levels. All models struggle with the hierarchical concepts, due to the highly non-identifiable nature of the problem, but ours manages to do better over a wider range (Fig. 3b). The hybrid geometry is most challenging for all models (Fig. 3c), presumably as they struggle to find the unique Schur-independent components without the benefit of the projection steps of Algorithm 1. We note in the case of the tree-structured data that hamming distance may be deceptive, since the concepts are very sparse and so any other sparse concept will be close to it; hence we also show the CKA. In terms of time, we are comparable but slightly faster with the other autoencoders.

## 4   APPLICATION TO NEURAL WORD EMBEDDINGS

Finally, we will see how this works on a simple example from word2vec (Mikolov et al., 2013), a well-known word embedding. The embedding is generated by predicting word-context pairs, in which contexts are the neighboring words in the Google News corpus. The embeddings can themselves be seen as a matrix factorization (Levy and Goldberg, 2014). What earned word2vec its notoriety is the fact that certain abstract concepts are encoded along parallel dimensions–exactly the kind of structure that we are looking for.

Since word2vec has embeddings for several million words, we need to select a subset to study. WordNet (Fellbaum, 1998) contains manually-encoded 'is-a' relations (i.e. 'dog' is-a 'canine') over a large vocabulary, and we used it to generate a list of words that are descendants of 'person'. This comes out to 3841 words. With around 4000 points in 300 dimensions, this has an intermediate dimensionality, and thus is a good use case for the hybrid sampling and refinement algorithm. Doing this we achieve a CKA of around 0.94 (Fig. 4a) using 993 concepts. The model splits into sparse and dense concepts (Fig.4b), which is indicative of an underlying low-dimensional structure with some high-dimensional deviations. It is the low-dimensional variables we will focus on interpreting.

We have replaced a high-dimensional continuous embedding with a high-dimensional binary embedding, and some additional tools are required to make sense of it. This is a problem for most large-scale interpretability methods, and modern strategies for analysis of word embeddings include asking large language models to say what all words in a cluster have in common (Huben et al., 2024).While there are also practical uses for binary embeddings (Andoni and Indyk, 2006; Koh et al., 2020), here we will showcase two styles of analysis which relate to our goals of abstraction and interpretability.

**Visualising global structure**   Our first approach will be to inspect two concepts across the whole dataset and to find the common variable across them. When we choose two concepts, this groups together the data into four clusters, based on which combination of values the word takes (Fig. 4c). We can then compute the "parallelism" of these concepts by measuring the angle that the vectors connecting centroids of each cluster make (Fig. 4d). For instance, we take the difference between clusters 1 and 2, which differ only by concept 1, and compare that to the difference between clusters 3 and 4 which also differ only by concept 1. Likewise for 1/3 and 2/4. Doing this for all pairs of discovered concepts, we pick the pair with the highest average parallelism (in our case, 0.92 and 0.90). We then project

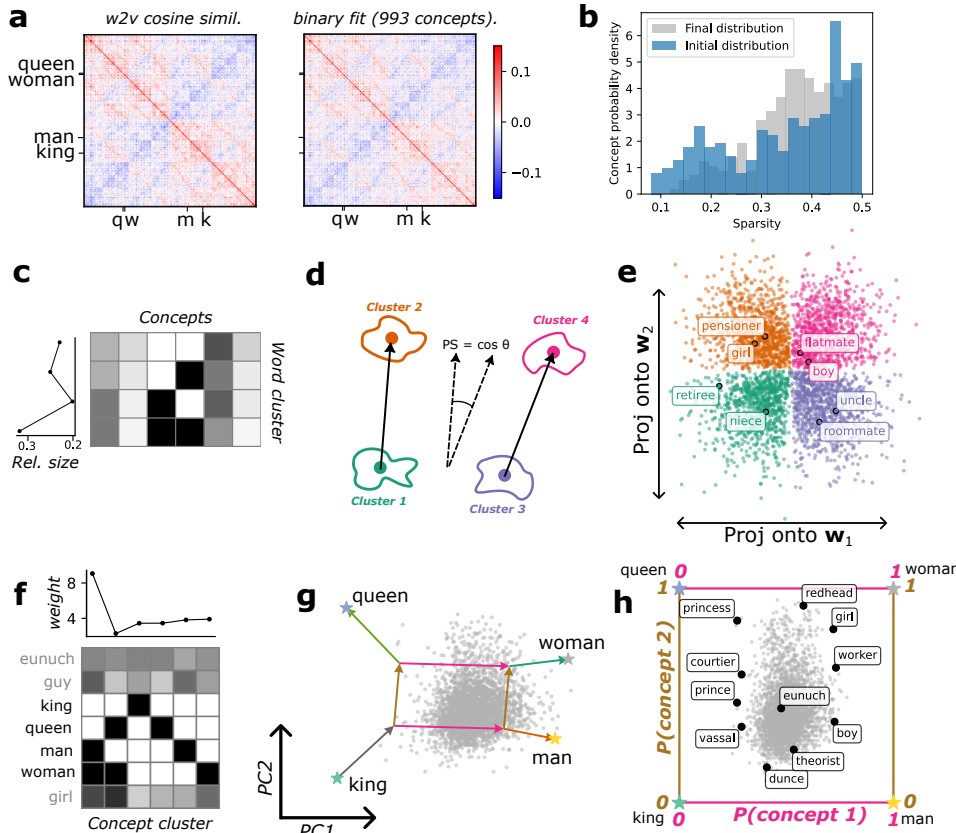

Figure 4: (**a**) Cosine similarity of 3841 words, in the word2vec representation (left) and the binary reconstruction (right). Words were selected using WordNet by taking all hyponyms of 'person' which were also present in word2vec. (**b**) The distribution of concept sparsity after fitting (blue) compared with the random projection initialization (grey). (**c**) Averages of the rows of **S** grouped according to two concepts (the middle ones). The relative size of each word cluster is indicated on the left plot. (**d**) A cartoon illustrating the parallelism score (PS) computation on four clusters. (**e**) With the two most parallel concepts, we project each word vector onto the corresponding columns of **W**, coloured according to its cluster. Sample words are highlighted. (**f**) PCA plot of four words of interest, with analogram overlaid and projection of all other selected words. (**g**) Example of sub-concept vectors, which are averages over concept vectors conditioned some subset of points. (**h**) The 'conceptual projection' of all fitted words onto the two high-level concepts. Highlighted words were selected by manual inspection.

all the data onto the features, **W**, associated with these two concepts, and find that the four clusters are well-separated (Fig. 4e).

To interpret the chosen concepts, we will see if there are consistent changes between pairs of words from one side of a concept to the other. We search for quadruplets which look the most like a square in the binary space, i.e. which have the fewest bits changing amongst them, and highlight the top two in Fig.4e. Going from the lower left to upper left cluster, we go from a familial term ("niece") to a generic term with the same gender ("girl"); we also go from a word more common in American English ("retiree") to its synonym in British English ("pensioner"). While it is not clear what these changes have in common with each other, the same types of changes are observed when moving up on the right two clusters ("uncle" to "boy", "roommate" to "flatmate"). It is somewhat harder to parse what changes from left to right. In one quadruplet, gender changes ("girl"/"boy" to "niece"/"uncle"), while it is hard to summarise what changes between "retiree" and "roommate". This suggests that

our discovered concepts may be too abstract, and further analysis is required to disentangle the more specific variables changing across each dichotomy.

**Visualising local structure** Instead of looking across the whole dataset, where a lot of things might be changing, we can focus on a particular quadruplet and try to interpret the concepts which separate them based on how other words vary along them. We will pick the famous quadruplet of "king", "queen", "man", and "woman". Conditioning on these four rows of $\mathbf{S}$, we find 6 kinds of concept (Fig. 4f), which can be visualised with the analogram in Fig .4g. From this we can see idiosyncratic concepts which group together single words, as well as two concepts which group together "king" and "queen", or "king" and "man".

While the "king"/"man" and "king"/"queen" concepts are often interpreted as "class" and "gender", respectively, it is not clear from just 4 words whether this is the right interpretation. When we take a subset, we are combining multiple dataset-level concepts into a single quadruplet-level concept; that is, there are many dataset-level concepts which group together "king" and "queen" vs. "man" and "woman". For example, "man" and "woman" are very generic terms, and something like 'specificity' is a concept which could plausibly affect the contexts in which they appear in the news. From the analysis before, it is clear that discovered concepts may sometimes correspond multiple variables.

To better understand the quadruplet-level concepts, we can see how the rest of the dataset varies along them on average. In Fig. 4e we are plotting this average value (which is between 0 and 1) for every word in grey, and some specific words are highlighted. A word's value along the first axis can be interpreted as "$x\%$ like 'man'-'king''', while the second axis "$y\%$ like 'queen'-'king'''. The resulting picture is certainly related to the geometry of the PCA plot, but it is not exactly the same and is more quantitatively interpretable. This kind of plot can complement the categorical, global-level analysis from before.

In general, the right side of concept 1 seems to be courtly roles rather than rulers *per se*, while the left side are perhaps called 'mundane'. Concept 2 does seem to capture some gendered words ('prince' vs 'princess' and 'boy' vs 'girl'), and the expected parallelisms are visible in this projection. Meanwhile, there are words which are not gendered but may nevertheless appear more often in gendered contexts in the news dataset, like "dunce" and "readhead". At the center of both concepts is "eunuch". We note that this qualitative picture persists for many random seeds and hyperparameter choices (annealing schedule, regularization, number of concepts).

## 5 Discussion

Here we studied the problem of turning a continuous representation into a logical one. We provided two simple algorithms with complementary use cases and demonstrate their efficacy. When dealing with very low-rank data, we leveraged results from previous work to develop a very fast and fairly robust method based rejection sampling. For when the data is higher-dimensional, we use a coordinate descent approach to refine the solution. Both algorithms enjoy convergence guarantees by connection to Hopfield networks, and thus represent a new application of that neuroscience model to optimization (Hopfield and Tank, 1985). We show favorable results on synthetic data when compared to gradient descent, and demonstrate several secondary analyses on word embeddings.

To the extent that we, humans, are engaged in concept discovery, our work here could also provide a minimalistic model of that cognitive process. We are not modeling the difficult process of discovering concepts from the real world: our model is linear, and therefore assumes a representation which already, at least approximately, encodes the relevant concepts. Instead, we model the process of turning a distributed, noisy code into abstract symbols and structures (Kemp and Tenenbaum, 2008). At some point, distributed activity in the brain is turned into discrete words, which is all that is heard by others. Future work can address how the inherent challenges of unsupervised abstraction, which we laid out, might be dealt with by more sophisticated algorithms with different inductive biases.

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

# A APPENDIX

## A.1 CONNECTIONS TO PCA AND CLUSTERING

Like many matrix factorizations, our method has an aesthetic similarity to the eigenvalue decomposition and PCA. In fact, when the concepts are uncorrelated with each other, i.e. they are orthogonal after subtracting the means, then it is actually equivalent to PCA. However–the rarity of this situation aside–PCA is not likely to find binary features when there is approximate rotation-invariance (geometric multiplicity) to the solution. Furthermore, when there are multiple factorizations, the sparsest one will have correlated features in general and thus not be recoverable.

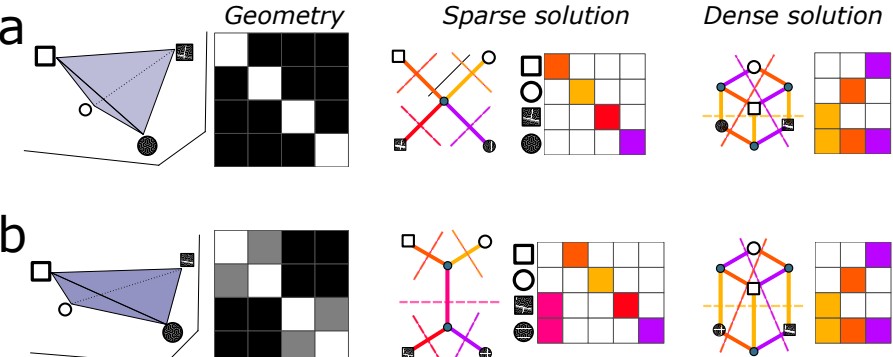

Figure 5: Examples of degenerate solutions. In row (**a**), we show the two possible embeddings of a tetrahedron geometry. **S** matrices are shown next to the 'analograms' with corresponding edges colored. We can get the columns of **S** by cutting the graph (removing edges) along the dashed color-coded lines. Both graphs have the same pairwise distances. Row (**b**) is largely the same, except there is one concept ('blank' vs 'shaded') which is present in both embeddings–in this sense some embeddings are partially identifiable.

We can also view our problem as a case of non-disjoint similarity-based clustering. Dasgupta (2016) defined an objective similar to ours for hierarchical clustering, and some of the theoretical results therein could be of interest. Work has also been done on generalizing $k$-means clustering to allow for overlap (Cleuziou, 2007; Whang et al., 2015) which is conceptually similar but uses a different cost function with a different implied generative model. The mixed-membership stochastic blockmodel (Airoldi et al., 2008) is another famous clustering model tackling a similar problem, and the connection to ours is made explicit by Sørensen et al. (2022). These clustering models generally incorporate constraints that we do not wish to include, or differ in their objective.

### A.2    ILLUSTRATIVE EXAMPLES

To build intuition for binary embeddings, we will go over some prototypical cases which can be inspected manually. We will imagine observations coming from three regular latent structures, and ask whether the ground truths are recoverable. In each diagram, the distances given are Euclidean, and the graph distances should match their square.

### A.2.1    INDEPENDENT CATEGORIES

This is the nicest case, one where we observe all possible combinations of the latent categories. The resulting geometry is a $d$-dimensional cube, far lower dimensionality than the number of observations ($2^d$):

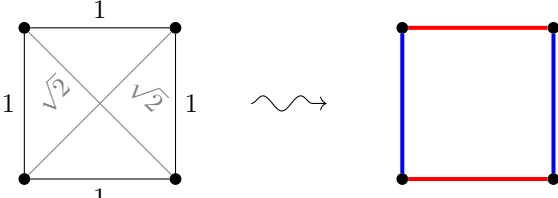

Above, I have drawn an example geometry for four points (on the left), and the corresponding graph of the embedding (on the right). In the graph, each edge is colored according to its corresponding category.

### A.2.2 MUTUALLY EXCLUSIVE CATEGORIES

This case is most similar to traditional clustering, in which categories cannot be combined. Every observation belongs to exactly one latent category, and the resulting dimensionality is potentially as high as the number of observations. The resulting geometry is one where every cluster of points is equidistant to every other.

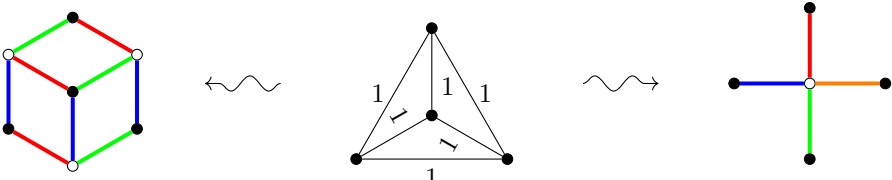

In our 4-item example above, there are two possible embeddings: a 3-dimensional one with more 'hidden' nodes (left) and a 4-dimensional one with fewer hidden nodes (right). The 4d one recapitulates our 'ground truth' latent structure since each item has its own category, but the 3d one does equally well. Intuitively, we might prefer the 4d solution since (1) it does not group together points unnecessarily and (2) there are fewer hidden nodes. Therefore, rather than the embedding dimension as the natural notion of parsimony, we suggest to use the number hidden nodes, or category size.

**Hierarchy** Instead of a flat clustering, imagine the data fall into hierarchical clusters in which some categories are subsets of others. The resulting geometry has intermediate dimensionality:

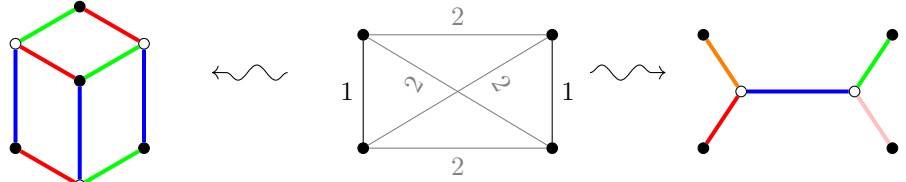

Notice this time that the graph on the left is isomorphic with the one in the previous example, but the blue edges have higher weight (i.e. they are longer in the drawing) to account for the larger distances. Instead, the graph on the right is a tree – something we would expect from hierarchical clustering. One category, the blue edges which separate the two pairs of closest points, appears in both solutions. So, at least some times, some categories might be identifiable even when the embedding as a whole is not.

### A.2.3 AN ORDINAL VARIABLE

Our observations now come from one ordered variable – for example, counting. The resulting geometry is a line. While all the previous examples admitted exact embeddings, this geometry is not embeddable on the cube and thus only has an optimal approximation. How can this be expressed in terms of categories?

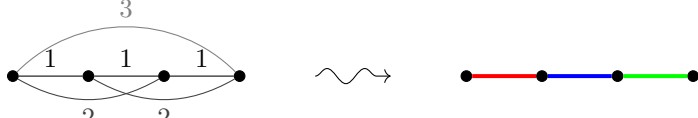

We end up with a graph of the expected topology, but the distances are slightly different since we are trying to match Hamming distances with squared Euclidean distances. With this example we can also see the utility and limits of our model: while we recover something meaningful at the graph level, we need 3 binary variables to describe a 1-dimensional struc-

ture[4]. So, while it is possible to get a sensible binary embedding of something fundamentally non-binary, it is not always efficient to do so.

### A.3 PARTIAL CUBES

Here is the sketch of the algorithm for recovering concepts from a partial cube graph: Start with an edge $e = (x, y)$, partition[5] the vertices into those which are closer to $x$ (call them $V_x$) or closer to $y$ (call them $V_y$). Each $V_x$ node gets a 0 label, each $V_y$ node gets a 1. Pick another edge, ignoring from now on any edges which cross the partition, and repeat the process. Here is an illustration of the process for a hexagon graph, coloring edged according to the partitions:

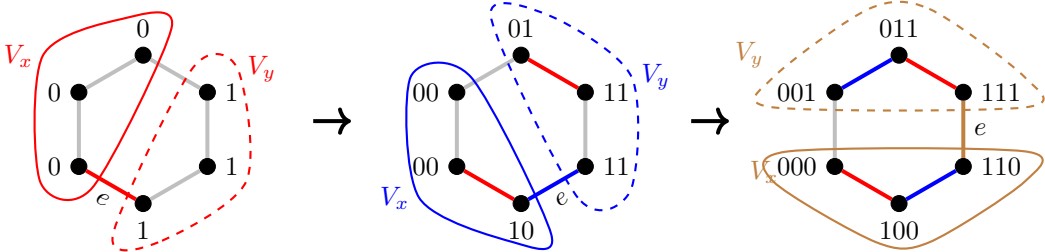

This algorithm is described in several places, but we used chapter 19 of Deza and Laurent (1997).

To go the other way around is hard, as mentioned in the main text. Our heuristic algorithm is based on two components: a global rule for finding hidden nodes, and a local rule for placing edges.

The global rule finds 'necessary' hidden nodes by taking the intersection of all differences between items. We will assume concepts are centered around the first item, such that the first row is all 0. We then take the intersection of all rows. For example, if items 1 and 2 differ by concepts $\{a, b, c\}$, while items 1 and 3 differ by $\{a, b, d\}$, then we will include the hidden node with concepts $\{a, b\}$. This is then repeated after re-centering around all other items.

The local rule is used to add edges to the graph, and ensure it is connected. It is a rule for which bits to add from a given node. Let's again assume we are centered around item 1. Construct a 'disjoint' matrix between concepts, which is equal to 1 only if $\mathbf{s}_\alpha^T \mathbf{s}_\beta = 0$. Then construct a 'superset' matrix, which is 1 only if $\mathbf{s}_\alpha^T \mathbf{s}_\beta = \mathbf{1}^T \mathbf{s}_\alpha$. The rule is that, at a given node $i$, with concepts $\mathbf{S}_i$, we are allowed to add any concepts which (1) are not disjoint with any concepts of $i$ and (2) have all their supersets active in $i$. For example, if I has the concepts ('dog', 'big'), then we could add 'great dane' but not 'pug', and not 'cat'.

By using the global rule for placing hidden nodes, and the local rule for laying down paths between all nodes, we can make a graph that is sometimes of manageable size. It works much better for sparser graphs like trees, and so a more picky method is going to be necessary moving forward.

### A.4 DERIVATION OF THE HOPFIELD ENERGY FUNCTION

Here we will show very explicitly how to convert the centered MSE loss function into a series of quadratic functions. When we use continuous variables, this will give us convex

---

[4]It is still possible to see the 1-dimensional structure by computing the 'lattice dimension' of the partial cube graph, which is the minimum dimension of an integer lattice that can contain the graph.

[5]This rule partitions the graph because partial cubes are bipartite. In fact, the partitioning is the basis of a certain binary relation, the Djokovic-Winkler relation, which is the theoretical basis of this construction.

problems. When we use binary variables, this will let us leverage heuristics for quadratic unconstrained binary optimization (QUBO), like the Hopfield network. We will break this down into a few parts: (1) high-level motivation for row-wise optimization; (2) showing how to deal with centering; and (3) writing down the quadratic functions.

Throughout, I will be working with matrices denoted by bold upper case letters. A corresponding lower case letter with a subscript means a row of the matrix, e.g. $\mathbf{s}_i$ is the $i^{\text{th}}$ row of $\mathbf{S}$. We will be using the convention that treats vectors as $p \times 1$ matrices, even when they are indexed as a row of a matrix.

In terms of dimension, the data is $\mathbf{X} \in \mathbb{R}^{p \times n}$ and the binary embedding is $\mathbf{S} \in \{0, 1\}^{p \times b}$. There is an equivalent version (which is what I originally did) of everything I am going to write in which $\mathbf{S} \in \{-1, 1\}^{p \times b}$, but this representation does not allow us to take advantage of sparsity so it is not what I use in the paper.

### A.4.1 MOTIVATION

Here we will go through every step of converting our loss to a series of quadratics. First consider a simpler version of our loss function, the mean squared error between the Gram matrices but *without any centering*. The derivation of the actual loss is messier, but will end up with something conceptually similar to this very simple thing.

$$\|\mathbf{X}\mathbf{X}^T - \mathbf{S}\mathbf{S}^T\|_F^2 = \operatorname{tr}(\mathbf{X}\mathbf{X}^T\mathbf{X}\mathbf{X}^T) + \operatorname{tr}(\mathbf{S}\mathbf{S}^T\mathbf{S}\mathbf{S}^T) - 2\operatorname{tr}(\mathbf{X}\mathbf{X}^T\mathbf{S}\mathbf{S}^T)$$

Remember that the trace terms, like those above, correspond to fourth-order summations in this case:

$$\operatorname{tr}(\mathbf{X}\mathbf{X}^T\mathbf{S}\mathbf{S}^T) = \sum_{i,j=1}^{p} \sum_{k=1}^{d} \sum_{l=1}^{b} \mathbf{X}_{ik}\mathbf{X}_{jk}\mathbf{S}_{il}\mathbf{S}_{jl}$$

and likewise for the other term. If we are minimizing with respect to all elements of $\mathbf{S}$ simultaneously, then we have a quartic (fourth-order) optimization, which is hard in general even for continuous variables. Instead, we can optimize one row at a time keeping all others fixed – a kind of block coordinate descent. By nudging one summation and separating the $i = j$ case we can make things easier. For the first term we have:

$$
\begin{aligned}
\operatorname{tr}(\mathbf{S}\mathbf{S}^T\mathbf{S}\mathbf{S}^T) &= \sum_{i=1}^{p} \sum_{k,l=1}^{b} \mathbf{S}_{ik}\mathbf{S}_{il} \sum_{j=1}^{p} \mathbf{S}_{jk}\mathbf{S}_{jl} \\
&= \sum_{i=1}^{p} \sum_{k,l=1}^{b} \mathbf{S}_{ik}\mathbf{S}_{il} \left( \sum_{j\neq i}^{p} \mathbf{S}_{jk}\mathbf{S}_{jl} + \mathbf{S}_{ik}\mathbf{S}_{il} \right) \\
&= \sum_{i=1}^{p} \sum_{k,l=1}^{b} \mathbf{S}_{ik}\mathbf{S}_{il} \sum_{j\neq i}^{p} \mathbf{S}_{jk}\mathbf{S}_{jl} + (\mathbf{S}_{ik}\mathbf{S}_{il})^2 \\
&= \sum_{i=1}^{p} \sum_{k,l=1}^{b} \mathbf{S}_{ik}\mathbf{S}_{il} \sum_{j\neq i}^{p} \mathbf{S}_{jk}\mathbf{S}_{jl} + \mathbf{S}_{ik}\mathbf{S}_{il} \\
&= \sum_{i=1}^{p} \mathbf{s}_i^T (\tilde{\mathbf{S}}^T \tilde{\mathbf{S}}) \mathbf{s}_i + (\mathbf{1}^T \mathbf{s}_i)^2
\end{aligned}
\tag{6}
$$

where we've used the fact the $0^2 = 0$ and $1^2 = 1$. In the last line, we're using $\tilde{\mathbf{S}}$ to indicate all the rows except $i$. For the second term of the loss it looks like:

$$
\begin{aligned}
\mathrm{tr}(\mathbf{X}\mathbf{X}^T\mathbf{S}\mathbf{S}^T) &= \sum_{i=1}^{p}\sum_{k=1}^{d}\sum_{l=1}^{b}\mathbf{X}_{ik}\mathbf{S}_{il}\sum_{j=1}^{p}\mathbf{X}_{jk}\mathbf{S}_{jl}\mathbf{X}_{ik}\mathbf{X}_{jk}\mathbf{S}_{il}\mathbf{S}_{jl} \\
&= \sum_{i=1}^{p}\sum_{k=1}^{d}\sum_{l=1}^{b}\mathbf{X}_{ik}\mathbf{S}_{il}\left(\sum_{j\neq i}^{p}\mathbf{X}_{jk}\mathbf{S}_{jl} + \mathbf{X}_{ik}\mathbf{S}_{il}\right) \\
&= \sum_{i=1}^{p}\sum_{k=1}^{d}\sum_{l=1}^{b}\mathbf{X}_{ik}\mathbf{S}_{il}\sum_{j\neq i}^{p}\mathbf{X}_{jk}\mathbf{S}_{jl} + (\mathbf{X}_{ik}\mathbf{S}_{il})^2 \\
&= \sum_{i=1}^{p}\mathbf{x}_i^T(\tilde{\mathbf{X}}^T\tilde{\mathbf{S}})\mathbf{s}_i + \mathbf{x}_i^T\mathbf{x}_i\mathbf{1}^T\mathbf{s}_i
\end{aligned}
\tag{7}
$$

which is quite similar as before, but linear in $\mathbf{s}_i$. This is all to show that, even though the MSE between Gram matrices is quartic, the updates for individual rows is quadratic (when one of them is binary).

### A.4.2 Centering

For theoretical Cortes et al. (2012); Sejdinovic et al. (2013) and empirical reasons, we want to use the centered distance/alignment. We can assume that the data is centered, but centering the binary matrix requires some consideration.

**Review** First, we review some facts about centering and constrained optimization. To center data, we remove mean from each column: $\bar{\mathbf{X}}_{ij} = \mathbf{X}_{ij} - \frac{1}{p}\sum_k \mathbf{X}_{kj}$. We can encode this transformation in a projection matrix, $\mathbf{H} = \mathbb{I} - \frac{1}{p}\mathbf{1}\mathbf{1}^T$, so that $\bar{\mathbf{X}} = \mathbf{H}\mathbf{X}$. The resulting kernel matrix, $\bar{\mathbf{K}} = \bar{\mathbf{X}}\bar{\mathbf{X}}^T$ is

$$
\begin{aligned}
\bar{\mathbf{K}}_{ij} &= (\mathbf{H}\mathbf{X}\mathbf{X}^T\mathbf{H})_{ij} \\
&= (\mathbf{H}\mathbf{K}\mathbf{H})_{ij} \\
&= \mathbf{K}_{ij} - \frac{1}{p}\sum_k \mathbf{K}_{ik} - \frac{1}{p}\sum_l \mathbf{K}_{lj} + \frac{1}{p^2}\sum_{kl}\mathbf{K}_{kl}
\end{aligned}
$$

where, remember, $p$ is the number of rows of $\mathbf{X}$. This centering transformation works on any kernel (symmetric positive semidefinite) matrix. One result is that the sum along each row and column of $\mathbf{K}$, and therefore so is the total sum. Another fun fact about $\mathbf{H}$ is that, being a projection, it is idempotent, meaning $\mathbf{H}\mathbf{H} = \mathbf{H}$.

Our objective function, as mentioned in the main text, is the centered kernel alignment. This is the Frobenius inner product of the centered kernel matrices, i.e.

$$
\begin{aligned}
\langle \mathbf{H}\mathbf{K}\mathbf{H}, \mathbf{H}\mathbf{Q}\mathbf{H}\rangle_F &= \mathrm{tr}(\mathbf{H}\mathbf{K}\mathbf{H}\mathbf{H}\mathbf{Q}\mathbf{H}) \\
&= \mathrm{tr}(\mathbf{H}\mathbf{H}\mathbf{K}\mathbf{H}\mathbf{H}\mathbf{Q}) \\
&= \mathrm{tr}(\mathbf{H}\mathbf{K}\mathbf{H}\mathbf{Q})
\end{aligned}
$$

where we've used the fact that $\mathrm{tr}(\mathbf{X}\mathbf{Y}\mathbf{Z}) = \mathrm{tr}(\mathbf{Z}\mathbf{X}\mathbf{Y})$ and the idempotence of $\mathbf{H}$ to unclutter the expressions. One implication of the simplified expression above, is that only one matrix actually needs to be centered. In these terms our objective, the CKA, is:

$$
\mathrm{CKA}(\mathbf{K}, \mathbf{Q}) = \frac{\mathrm{tr}(\mathbf{H}\mathbf{K}\mathbf{H}\mathbf{Q})}{\sqrt{\mathrm{tr}(\mathbf{H}\mathbf{K}\mathbf{H}\mathbf{K})\,\mathrm{tr}(\mathbf{H}\mathbf{Q}\mathbf{H}\mathbf{Q})}}
$$

Since the above has a square root and a fraction, we would rather use a different form for optimization. The CKA is analogous to the cosine similarity of two real vectors, and so one can imagine that maximizing the cosine–equivalent to minimizing the angle between the vectors–could be equivalent to minimizing the distance. In the context of constrained

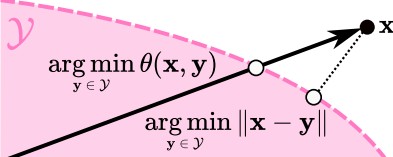

Figure 6: Example of the inequivalence of cosine and distance in constrained optimization.

optimization, that is not always the case (see Fig. 6). One way to make them equivalent, though, is to ensure that our variable can be freely scaled.

Just for fun, even though it's kind of obvious, let's show why scaling makes the two equivalent. We can add a non-negative scaling variable, $\sigma$, and pick the minimum distance with respect to that:

$$
\begin{aligned}
\mathcal{L}(\mathbf{y}) &= \min_{\sigma \geq 0} \|\mathbf{x} - \sigma \mathbf{y}\|^2 \\
&= \min_{\sigma \geq 0} \langle \mathbf{x}, \mathbf{x} \rangle - 2\sigma \langle \mathbf{x}, \mathbf{y} \rangle + \sigma^2 \langle \mathbf{y}, \mathbf{y} \rangle \\
&= \langle \mathbf{x}, \mathbf{x} \rangle - 2 \frac{\langle \mathbf{x}, \mathbf{y} \rangle}{\langle \mathbf{y}, \mathbf{y} \rangle} \langle \mathbf{x}, \mathbf{y} \rangle + \frac{\langle \mathbf{x}, \mathbf{y} \rangle^2}{\langle \mathbf{y}, \mathbf{y} \rangle^2} \langle \mathbf{y}, \mathbf{y} \rangle \\
&= \langle \mathbf{x}, \mathbf{x} \rangle - \frac{\langle \mathbf{x}, \mathbf{y} \rangle^2}{\langle \mathbf{y}, \mathbf{y} \rangle}
\end{aligned}
$$

which we could do since the second line is just a single-variable quadratic equation. It is a constrained minimization, but since the Frobenius inner product is non-negative, the unconstrained $\sigma^* \geq 0$ so we can just plug it in. Notice that minimising this is the same as maximising the cosine:

$$
\begin{aligned}
\arg\min_{\mathbf{y}} \langle \mathbf{x}, \mathbf{x} \rangle - \frac{\langle \mathbf{x}, \mathbf{y} \rangle^2}{\langle \mathbf{y}, \mathbf{y} \rangle} &= \arg\min_{\mathbf{y}} 1 - \frac{\langle \mathbf{x}, \mathbf{y} \rangle^2}{\langle \mathbf{y}, \mathbf{y} \rangle \langle \mathbf{x}, \mathbf{x} \rangle} \\
&= \arg\max_{\mathbf{y}} \frac{\langle \mathbf{x}, \mathbf{y} \rangle^2}{\langle \mathbf{y}, \mathbf{y} \rangle \langle \mathbf{x}, \mathbf{x} \rangle} - 1 \\
&= \arg\max_{\mathbf{y}} \frac{\langle \mathbf{x}, \mathbf{y} \rangle}{\sqrt{\langle \mathbf{y}, \mathbf{y} \rangle \langle \mathbf{x}, \mathbf{x} \rangle}}
\end{aligned}
$$

Hopefully we are fully convinced that maximising alignment is the same as minimising distance so long as we can freely scale.

**Application**  In our model, $\mathbf{S}$ can be freely scaled by the $\pi$ values, or more cheekily when $b$ is unbounded, by just having many duplicates of certain concept vectors. For that reason, we will assume that scale is not an issue and comfortably try to minimise the distance between the centered kernels:

$$
\|\bar{\mathbf{X}}\bar{\mathbf{X}}^T - \bar{\mathbf{S}}\bar{\mathbf{S}}^T\|_F^2 = \text{tr}(\mathbf{HSS}^T\mathbf{HSS}^T) - 2\,\text{tr}(\mathbf{HXX}^T\mathbf{HSS}^T) + \text{const}(\mathbf{S})
$$

For simplicity, we will drop the $\pi$, but it's not hard to add back later. We can use the row-wise updates from the before, but the centering requires some slight tweaking since each row contributes to the mean. We will re-frame everything in terms of the recursive alignment, and find that it makes things easier.

Let's say we have only seen $p$ items, so that $\mathbf{X}$ and $\mathbf{S}$ have $p$ rows. We can compute the centered distance for that. We will now show how to update the loss when a new row is added to each matrix. The kernels when we get row $p + 1$ are:

$$
\mathbf{K}^{(p+1)} = \begin{pmatrix} \mathbf{K}^{(p)} & \mathbf{k} \\ \mathbf{k}^T & k_0 \end{pmatrix}, \quad \mathbf{Q}^{(p+1)} = \begin{pmatrix} \mathbf{Q}^{(p)} & \mathbf{q} \\ \mathbf{q}^T & q_0 \end{pmatrix}
$$

Furthermore, let's assume that $\mathbf{K}^{(p)}$ and $\mathbf{Q}^{(p)}$ are already centered. We will say that the row-mean of $\mathbf{S}$ is $\langle \mathbf{s} \rangle = \frac{1}{p}\mathbf{S}^T\mathbf{1}$, so that $\mathbf{Q}^{(p)} = (\mathbf{S} - \mathbf{1}\langle \mathbf{s} \rangle^T)(\mathbf{S}^T - \langle \mathbf{s} \rangle \mathbf{1}^T)$. Likewise for $\mathbf{X}$ and

$\mathbf{K}^{(p)}$. The appendages are thus:

$$\mathbf{q} = (\mathbf{S} - \mathbf{1}\langle\mathbf{s}\rangle^T)(\mathbf{s} - \langle\mathbf{s}\rangle)$$

$$q_0 = (\mathbf{s} - \langle\mathbf{s}\rangle)^T(\mathbf{s} - \langle\mathbf{s}\rangle)$$

where $\mathbf{s}$ is the new row of $\mathbf{S}$.

This all makes centering $\mathbf{K}^{(p+1)}$ and $\mathbf{Q}^{(p+1)}$ straightforward. Here it is after some simplification:

$$\bar{\mathbf{Q}}_{ij}^{(p+1)} = \begin{cases} \mathbf{Q}_{ij}^{(p)} - \frac{1}{p+1}\mathbf{q}_i - \frac{1}{p+1}\mathbf{q}_j + \frac{1}{(p+1)^2}q_0 & i,j = 1,..p \\ \frac{p}{p+1}\mathbf{q}_i - \frac{p}{(p+1)^2}q_0 & i = 1,...p, j = p+1 \\ \frac{p^2}{(p+1)^2}q_0 & i = p+1, j = p+1 \end{cases}$$

which just comes from the fact that $\mathbf{Q1} = 0$ and $\mathbf{1}^T\mathbf{q} = 0$. The same can be done for $\mathbf{K}$ of course.

Without replicating the algebra here, we can use this form to compute the alignment of $\bar{\mathbf{Q}}_{(p+1)}$ and $\bar{\mathbf{K}}^{(p+1)}$:

$$\left\langle \bar{\mathbf{Q}}^{(p+1)}, \bar{\mathbf{K}}^{(p+1)} \right\rangle_F = \left\langle \bar{\mathbf{Q}}^{(p)}, \bar{\mathbf{K}}^{(p)} \right\rangle_F + 2t\mathbf{k}^T\mathbf{q} + t^2 k_0 q_0 \tag{8}$$

in which we've defined $t = \frac{p}{p+1}$. The same update can be used for the other inner products, to give an update of the loss. Plugging in the form of $\mathbf{q}$ and $q_0$, we've shown how to write the loss with respect to one row of $\mathbf{S}$ in a way that is quadratic in that row.

### A.4.3 DERIVATION

All that remains is to write the row-wise loss out explicitly. The form of $\mathbf{q}$ that we supplied earlier is not the only one, and in fact we've considered a few different ways to extend the model kernel. For example, in section A.5 we consider an 'infinite-dimensional' version, in which $\mathbf{q}$ is updated using variables in $[0, 1]$. In principle, especially if you aren't committed to a quadratic loss, $\mathbf{q}$ and $\mathbf{k}$ could be formed by many functions. Here we will stick with the simple version given above.

We will be plugging in the equation for $\mathbf{q}$ into the recursive form of the inner products (8) of the loss:

$$\left\|\bar{\mathbf{Q}}^{(n+1)} - \bar{\mathbf{K}}^{(n+1)}\right\|_F^2 = \left\langle \bar{\mathbf{Q}}^{(n+1)}, \bar{\mathbf{Q}}^{(n+1)} \right\rangle_F + \left\langle \bar{\mathbf{K}}^{(n+1)}, \bar{\mathbf{K}}^{(n+1)} \right\rangle_F - 2\left\langle \bar{\mathbf{Q}}^{(n+1)}, \bar{\mathbf{K}}^{(n+1)} \right\rangle_F$$

$$= \left\|\bar{\mathbf{Q}}^{(n)} - \bar{\mathbf{K}}^{(n)}\right\|_F^2 + 2t\mathbf{q}^T\mathbf{q} + t^2 q_0^2 - 4t\mathbf{k}^T\mathbf{q} - 2t^2 k_0 q_0 + \text{const}(\mathbf{q})$$

That is, we'll be plugging $\mathbf{q} = \bar{\mathbf{S}}(\mathbf{s} - \langle\mathbf{s}\rangle)$ and $q_0 = (\mathbf{s} - \langle\mathbf{s}\rangle)^T(\mathbf{s} - \langle\mathbf{s}\rangle)$ into the equation above. We will end up with something similar to the uncentered forms (6 and 7) but too messy to reproduce at this hour. After gathering terms, we have:

$$\mathcal{L}(\mathbf{s}) = \mathbf{s}^T\mathbf{J}\mathbf{s} - 2\mathbf{h}^T\mathbf{s}$$

$$\mathbf{J} = 2\bar{\mathbf{S}}^T\bar{\mathbf{S}} + t\langle\tilde{\mathbf{s}}\rangle\langle\tilde{\mathbf{s}}\rangle^T \tag{9}$$

$$\mathbf{h} = \mathbf{J}\langle\mathbf{s}\rangle + t\langle 1 - \mathbf{s}\rangle^T\langle\mathbf{s}\rangle\langle\tilde{\mathbf{s}}\rangle - tk_0\langle\tilde{\mathbf{s}}\rangle + 2\mathbf{S}^T\mathbf{k} \tag{10}$$

$$\langle\tilde{\mathbf{s}}\rangle = 2\langle\mathbf{s}\rangle - \mathbf{1}$$

which is a quadratic unconstrained binary objective in $\mathbf{s}$, as promised. Note that we are using $\bar{\mathbf{S}} \doteq \mathbf{S} - \mathbf{1}\langle\mathbf{s}\rangle^T$ in the expressions above. They can also be rewritten in terms of the uncentered $\mathbf{S}$ and some rank-one terms, which allows us to take advantage of the sparsity of $\mathbf{S}$ during optimization.

**Note on asymmetric case**  While we chose the kernel alignment objective for pragmatic reasons, having a locally-implemented (biologiclly plausible) algorithm actually relies on it. If we minimize the mean squared error with alternating least squares (alternating between optimization of $\mathbf{W}$ and $\mathbf{S}$), then updates for $\mathbf{S}$ are also row-wise quadratic functions. But

if we are using a Hopfield network to do the optimization, then the weights would be something like $\mathbf{W}^T\mathbf{W}$, which would require neurons to correlate their efferent weights in order to compute their recurrent weights, instead of correlating their activity. The asymmetric factorization, $\mathbf{SW}^T$ seems a bit trickier to optimize according to Sørensen et al. (2022) and Kolomvakis and Gillis (2023), but there may still be some advantages to it (like speed, and more flexible loss functions) that are worth tricky optimization, and giving up the pretty Hopfield algorithm.

## A.5 Convex relaxation

---

**Algorithm 3** Convex item assignment

---

1: **function** FIT($\mathbf{X}$, $K \in \mathbb{N}$, $n \leq p$)
2:     **for** $k = 1, .., K$ **do** ▷ *Parallel branches*
3:         $\mathbf{S}(k) = \{0,1\}^n$
4:         $\boldsymbol{\pi}(k) = \arg\min_{\boldsymbol{\pi}} 3$
5:         **for** $i = n, ..., N$ **do**
6:             $\varepsilon \sim \mathcal{N}(0, \mathbb{I})$
7:             $\hat{\mathbf{p}}, \pi_0 = \arg\min 12 + \varepsilon^T\hat{\mathbf{p}}$
8:             $\mathbf{S}(k) \leftarrow \begin{pmatrix} \mathbf{S} & \mathbf{S} & \mathbf{0}_n \\ \mathbf{1}_b^T & \mathbf{0}_b^T & 1 \end{pmatrix}$
9:             $\boldsymbol{\pi}(k) \leftarrow (\hat{\mathbf{p}}, \boldsymbol{\pi} - \hat{\mathbf{p}}, \pi_0)$
10:     **Return** $\mathbf{S}$, $\boldsymbol{\pi}$

---

Here we provide a continuous relaxation of the discrete row updates. 3.

We assume we have a fit for the first $p$ items. The kernels when we see item $p+1$ are:

$$\mathbf{K}^{(p+1)} = \begin{pmatrix} \mathbf{K}^{(p)} & \mathbf{k} \\ \mathbf{k}^T & k_0 \end{pmatrix}, \quad \mathbf{Q}^{(p+1)} = \begin{pmatrix} \mathbf{Q}^{(p)} & \mathbf{q} \\ \mathbf{q}^T & q_0 \end{pmatrix}$$

The loss we want to compute is

$$\left\| \bar{\mathbf{Q}}^{(p+1)}, \bar{\mathbf{K}}^{(p+1)} \right\|_F = \left\langle \bar{\mathbf{Q}}^{(p+1)}, \bar{\mathbf{Q}}^{(p+1)} \right\rangle_F + \left\langle \bar{\mathbf{K}}^{(p+1)}, \bar{\mathbf{K}}^{(p+1)} \right\rangle_F - 2 \left\langle \bar{\mathbf{Q}}^{(p+1)}, \bar{\mathbf{K}}^{(p+1)} \right\rangle_F$$

To compute the required inner products, we need to double-center the matrices. Recall from section A.4 that the recursive update of the loss given $\mathbf{k}$, $k_0$, $\mathbf{q}$, and $q_0$ is:

$$\left\langle \bar{\mathbf{Q}}^{(p+1)}, \bar{\mathbf{K}}^{(p+1)} \right\rangle_{\mathbf{H}} = \left\langle \bar{\mathbf{Q}}^{(p)}, \bar{\mathbf{K}}^{(p)} \right\rangle_{\mathbf{H}} + 2t\bar{\mathbf{k}}^T\bar{\mathbf{q}} + t^2\bar{k}_0\bar{q}_0$$

where I have abbreviated $t = \frac{n}{n+1}$. Plugging these into the loss equation and fiddling a bit, we have:

$$\|\mathbf{K}^{(p+1)} - \mathbf{Q}^{(p+1)}\|_{\mathbf{H}}^2 = \|\mathbf{K}^{(p)} - \mathbf{Q}^{(p)}\|_{\mathbf{H}}^2 + 2t\bar{\mathbf{q}}^T\bar{\mathbf{q}} + t^2\bar{q}_0^2 - 2\left(2t\bar{\mathbf{k}}^T\bar{\mathbf{q}} + t^2\bar{k}_0\bar{q}_0\right) + 2t\bar{\mathbf{k}}^T\bar{\mathbf{k}} + t^2\bar{k}_0^2$$

$$= \|\mathbf{K}^{(p)} - \mathbf{Q}^{(p)}\|_{\mathbf{H}}^2 + 2t\left\|\bar{\mathbf{q}} - \bar{\mathbf{k}}\right\|_2^2 + t^2(\bar{q}_0 - \bar{k}_0)^2 + \text{const.} \tag{11}$$

which we can get by completing the square. This means that we are just solving a (slightly) weighted least squares between the centered data and prediction. So long as $\bar{\mathbf{q}}$ and $\bar{q}_0$ are linear functions of our parameters (which they will be), minimizing the loss at each step is just a quadratic program. We will now see that they are.

It remains to write out $\bar{\mathbf{q}}$ and $\bar{q}_0$. The centered form of our concept update is:

$$\bar{\mathbf{S}} \leftarrow \begin{pmatrix} \bar{\mathbf{S}} & \bar{\mathbf{S}} & \mathbf{0}_n \\ 1 - \langle \mathbf{s} \rangle & -\langle \mathbf{s} \rangle & 1 \end{pmatrix}$$

and remember that the $\boldsymbol{\pi}$ update is

$$\boldsymbol{\pi} \leftarrow (\boldsymbol{\Pi}\mathbf{p}, \boldsymbol{\Pi}(1 - \mathbf{p}), \pi_0)$$

To get the kernel values we need to take the dot product of the last row with the first $n$ rows and with itself (weighting columns by $\boldsymbol{\pi}$). Working through the algebra on that we get:

$$\bar{\mathbf{q}} = \bar{\mathbf{S}}\boldsymbol{\Pi}(\mathbf{p} - \langle \mathbf{s} \rangle)$$

and

$$\bar{q}_0 = (1 - 2\langle \mathbf{s} \rangle)^T \boldsymbol{\Pi}\mathbf{p} + \langle \mathbf{s} \rangle^T \boldsymbol{\Pi} \langle \mathbf{s} \rangle + \pi_0$$

which are indeed linear functions of our parameters, $\mathbf{p}$ and $\pi_0$.

The resulting quadratic program is:

$$\underset{\hat{\mathbf{p}}, \pi_0}{\arg\min} \quad 2 \left\| \bar{\mathbf{S}}(\hat{\mathbf{p}} - \langle \hat{\mathbf{s}} \rangle) - \mathbf{k} \right\|_2^2 + \frac{n}{n+1} \left( (\mathbf{1} - 2\langle \mathbf{s} \rangle)^T \hat{\mathbf{p}} + \pi_0 + \langle \hat{\mathbf{s}} \rangle^T \langle \mathbf{s} \rangle - k_0 \right)^2 \quad (12)$$

$$\text{s.t.} \quad \mathbf{0} \le \hat{\mathbf{p}} \le \boldsymbol{\pi}, \ 0 \le \pi_0$$

