# OpenReview forum: "Unsupervised Learning of Categorical Structure"
_ICLR.cc/2025/Conference — Submitted to ICLR 2025_

### Official Review · Reviewer_Q2e1 · 2024-10-30

**Soundness:** 2
**Presentation:** 2
**Contribution:** 2
**Rating:** 6
**Confidence:** 2

**Summary:**

This paper aims to extract binary representations from data, which preserve the latent structure inherent in original data. The authors set up an objective based on centered kernel alignment between the data and binary representations and propose two iterative algorithms for factorizing the data into binary components: one designed for low-rank data and another for full-rank data. They validate their approach with numerical experiments on simulated low-rank data, hybrid low-rank and hierarchical data, and a subset of WordNet, demonstrating the effectiveness of the proposed algorithms.

**Strengths:**

**S1.** Using the binary matrix factorization to identify geometrical representations seems novel.

**S2.** This paper proposes two algorithms to handle both low-rank and full-rank data, making this framework versatile for various data.

**S3.** This paper demonstrates the efficacy of the proposed methods through several numerical validations. Additionally, it includes analyses of extracted binary representations and an ablation study to test robustness in the presence of noise.

**Weaknesses:**

**W1.** The introduction could benefit from further refinement. For example, although the paper introduces two algorithms, there is no clear description of them in this section. Additionally, introducing the concept of binary representation in the third paragraph could help readers grasp the paper's focus more effectively.

**W2.** This paper argues that binary representations can improve the interpretability of neural representations. While this paper provides several analyses of representational quality, whether these representations truly capture the data's latent structure remains ambiguous.

**W3.** Providing a more detailed explanation of the transformation from Equation 2 to Equation 3 would assist readers in understanding the method.

**Questions:**

**Q1.** From the perspective of representational geometry and sparsity, previous work [1] may be related to this paper. Discussing the relationship between sparsity and representational geometry would provide valuable insights for readers.

**Q2.** In Section 2.2.1, the authors discuss the examples in Figure 3 but do not fully clarify the concept of a hidden node. Could they provide a more detailed explanation of this term?

**Q3.** I am curious whether these algorithms can be used in end-to-end training where each word representation is dynamically changed or if they are applicable solely for post-hoc analysis after training. I would appreciate it if the authors could share their thoughts on this.

[1] Elhage, et al., "Toy Models of Superposition", Transformer Circuits Thread, 2022.

---

> ### Author Response · Authors · 2024-12-02
> **Response to reviewer Q2e1**
>
> We thank the reviewer for the comments on our manuscript. In addition to the general changes and comments, we highlight changes which address the reviewer's concerns specifically in brown.
>
> (1) Regarding the introduction: We have reworded parts of the introduction (second paragraph) to more clearly motivate our desire for categorical latent variables, and also explicitly mention our contribution of efficient algorithms for SBMF in the final paragraph.
>
> (2) Regarding fidelity: While the recovery of true latent structure is difficult to assess in the case of static word embeddings, our synthetic experiments demonstrate that the algorithm is capable of recovering ground truth. In the case that a representation comes from a neural network or other system, causal interventions are available, and while interesting, we are not investigating that application in this paper.
>
> (3) Regarding Equation 3: We have replaced the optimization equation over pi with the centered MSE loss, which is more directly relevant to the binary version of the algorithm. A step-by-step equivalence between the two is provided in the Appendix as well.
>
> Questions:
>
> Q1: This is an important connection, and the basis of the sparse autoencoders that we now discuss directly in the improved prior work section. Given that we are mostly working with orthogonal encodings, we are actually assuming a different regime than superposition (which allows for correlated coding vectors), so in a sense we are offering a complementary lens for viewing representations.
>
> Q2: We have corrected the unexplained introduction of this concept, and now only reference the size of the analogram (which is explained in the prior section). To clarify what we meant before: a binary embedding can be viewed as a subset of a hypercube, i.e. data are identified with some vertices of the cube. There will be other vertices which do not correspond with the embedding of any data point. Some of these vertices will be `in between' the data points, meaning that a shortest path between the points must pass through them, and those are what we call hidden nodes.
>
> Q3: This is indeed an interesting application, and directly relevant to the concept bottleneck models that we now highlight in the introduction. In short, these models require that a network's output passes through a set of pre-specified concepts (binary nodes) in an effort to make the network more interpretable. Our method, given its speed, could potentially serve as an unsupervised concept layer, although we cannot anticipate how it would interact with the training procedure of a large network. Given that our model is just applying a rotation to the continuous representation, we would expect any benefits to be more in interpretability than in performance or generalization.

---

> > ### Comment · Reviewer_Q2e1 · 2024-12-03
> >
> > Thank you to the authors for their responses and for addressing my concerns.
> >
> > In the second paragraph of the introduction, the authors claim that continuous approaches suffer from “quantization errors.” However, I believe that deriving the latent structure based on binary representations may rather induce “quantization errors.” I would appreciate the authors sharing their perspectives on this point.

---

> > > ### Author Response · Authors · 2024-12-03
> > > **Reply by the authors to reviewer Q2e1**
> > >
> > > Thank you for the fast response. Our usage of this term is perhaps somewhat ambiguous and we would be happy to correct it in the camera ready.
> > >
> > > We agree than any discrete analysis, like clustering or our own method, could discard important information and thus be misleading. But we aim to recover discrete structure when it is present in the data, and so our notion of "error" is based on that in particular.
> > >
> > > Concretely, a continuous latent variable model (like a sparse autoencoder) will be a good model on the basis of its good reconstruction of the data, in our case quantified by distance preservation -- when the continuous latents are discretized, this will generally reduce the quality of the fit, i.e. the binarised latents no longer match the distances of the data. The goal of our method is to avoid the reconstruction errors from this process, which is done often in the field, by directly finding binary latents which preserve distances.

---

> > > > ### Comment · Reviewer_Q2e1 · 2024-12-03
> > > >
> > > > Thank you for the authors' clarification and for addressing my concerns. I have decided to increase the score.

---

### Official Review · Reviewer_aC8b · 2024-10-31

**Soundness:** 2
**Presentation:** 2
**Contribution:** 2
**Rating:** 5
**Confidence:** 2

**Summary:**

The authors present a representational decomposition technique to find bases of binary logical components. They describe how to do the technique. They then show that the method can work for a number of geometries in synthetic settings. And lastly they show that the method can be used with word2vec embeddings to some success.

**Strengths:**

- Decomposing representations into logical components maybe could be promising for generalization?
- The authors showed the technique in both toy settings and more practically relevant settings (word2vec)
- From what I understood, the theoretical backing was sound

**Weaknesses:**

- unclear what the author's contributions are. is it the introduction of a new factorization technique?
- unclear why we should care about this factorization technique. the paper needs more focus on relevance.
- the structure of the paper would benefit from more clarity on the high level goals of each section
- I found the majority of the technical details difficult to understand because it sometimes felt like it was unclear how the details fit into the greater picture

**Questions:**

- What problems do these logical variables solve? How can they be used?
    - is the suggestion that this is a process that occurs in biological systems?
    - do these embeddings somehow improve performance on something?
- There is a big data dependence issue when creating the kernel matrix
    - what dataset do you use here in practical settings?
    - how can you ensure you're using a relevant dataset for the model you are examining?
    - how can you ensure that the data sufficiently spans the model's representation space?
- how does method compare to something like locality sensitive hashing for forming discrete representations?
- why is it necessarily a bad thing that neural systems use continuous representations? In general, I see the opposite complaint more frequently, that artificial networks impose too much slot-like and discrete structure in ways that the human brain does not
- if you weight the logical variables, are they not still continuous representations?

---

> ### Author Response · Authors · 2024-12-02
> **Response to reviewer aC8b**
>
> We thank the reviewer for the comments on our manuscript
>
> (1) Regarding contribution: In response to the reviewer's concern, we have clarified our specific contributions in an improved introductory section. In short, we propose SBMF as a useful tool for constructing categorical representations, and provide algorithms which handle a broader set of cases than previous ones.
>
> (2) Regarding motivation: Our inclusion of more conceptually relevant prior work, and a new contribution section, can hopefully provide some clarity about the relevance of this work to open problems in the field, such as concept bottleneck models and hashing.
>
> (3) Regarding narrative structure: In our substantial revisions to the text, we have striven for more narrative clarity and consistency, especially in the word2vec section.
>
> (4) Regarding the bigger picture: Related to the previous points, our improved section on prior work now provides a more complete sense of the positioning of our work, and we return to these motivating examples in the improved word2vec section. We have also cleaned up the notation and unnecessary introduction of new symbols, as was pointed out by other reviewers.
>
> Q1: The variables are primarily for interpretation, essentially a non-disjoint clustering. However there are also potential practical uses, such as hashing (as the reviewer mentions) and in quasi-symbolic neural networks like concept bottleneck models.
>
> Q2: In general we can use any positive definite matrix, including the linear kernel of some representation (as we do with word2vec) or a non-linear kernel. We have applied the method successfully to neural data and co-occurrence matrices from , but did not have the space to include it. The extent to which our binary generative model is appropriate for the data can only be assessed by inspecting the quality of the fit, since it is hard to tell from the geometry alone whether it admits a clean factorization.
>
> Q3: This can indeed be seen as a method for locality sensitive hashing, and we now put greater emphasis on connection to prior work on this topic. Beyond its methodological differences, the hashing literature has a very different emphasis, and does not generally take the perspective of latent variable inference or assess the interpretability of the hashes. But it is possible our method would be nevertheless useful as a hashing tool.
>
> Q4: It is not at all a bad thing for neural systems to use continuous representations, as the reviewer points out, it is just difficult to interpret. Our method is tailored for the case where there is some discrete structure which is hidden by a rotation in neural space, of which we provide some examples from the literature in the introduction.
>
> Q5: The weighting of the variables allows us to capture a wider range of continuous geometries, but each variable is still binary in that it can take one of only two values -- rather than $\{0,1\}$, or $\{-1,1\}$, it can be $\{0,\pi\}$, or $\{-\pi,\pi\}$.

---

### Official Review · Reviewer_wFCT · 2024-11-08

**Soundness:** 3
**Presentation:** 4
**Contribution:** 3
**Rating:** 6
**Confidence:** 4

**Summary:**

This paper presents two related algorithims intended to guide unsupervised category learning. Overall the approach presented here tries to explain high-dimensional embeddings in terms of a linear combination of concept vectors. By taking a sets of concept vectors and embeddings then assigning concepts to each embedding via binary weights. This is done by computing a kernel matrix, comparing each embedding with all the others, and trying to assign categories to each embedding that preserves the similarity structure of the embedding space. That is; two embeddings that are similar should have similar category assignments. To do this process efficiently this work uses gradient based methods. In cases where the embeddings are low rank they search for category vectors that are present in the embedding using a hopfield network. Looking to identify which categorisations of the embeddings are structure preserving. This method is intractable in cases with higher-rank data, as a result they propose a second method that builds all category vectors at once, but looking at a single embedding at a time. By iterating through all the embeddings this iterativley builds category vectors that maximally preserve the similarity structure of the input space. Experiments on artificially generated data shows these methods are relatively good at recovering the undelying strucutre of the space. Additionally these methods are relatively fast when data is low rank with a small number of samples although neither algorithm scales terrifically well with time complexity past 1000 points becoming quite high. A final set of experiments applies these methods to word2vec embeddings, fitting binary concepts to a subset of word2vec, and discussing the potential semantic interpretation of different concepts via qualitative analysis.

**Strengths:**

Overall this paper is very well written, presenting work on an interesting topic, with well designed visuals. The text tries quite hard to give readers an intuition for why the authors pursue certain lines of enquiry before working through it. The introduction gives sufficient context for the approach, and builds some motivation for the line of work. Additionally experiments are presented on both ariticial and natural data. Allowing the reader to get a sense for exactly how accurate the method presented here is on data with a known underlying structure, before showing that the method still provides interesting results on more complex embeddings.

**Weaknesses:**

While I would like to be clear that overall I do think this paper is strong, with solid scientific merit, I have a couple of areas of concern. If these are adequately addressed I am open to raising my score.

A) The formalisations are in some places difficult to follow, with certain background literature being under-introduced.

In section 2, equation 2.2 Q is defined as yTy yet I don’t appear to see where y is defined. In other places the notion seems overly verbose and a bit too formal without arming the reader with a sufficiently strong intuition.  I had difficulty understanding equation 4 given the notation and inline explanation.

Terminologically, centered kernel alignment (CKA) seems central to this work but is mentioned in a single sentence without providing a high-level intuition before introducing it formally. Some discussion of making the similarity of one space align with the similarity of another before talking about Gram matrices would be welcome.

B) The authors discuss embeddings from 4 different perspectives, in terms of real-valued space, kernel space, binary concepts, and geometry. It would be helpful to either make clearer how these relate to each other, or to focus on fewer of them.

In the introduction they note: "These factors were analyzed discretely (is it active or not), but the continuous degrees of ‘active’ can make them a little harder to interpret them as categories or symbols. For that reason, we see advantage in having factors which are categorical by construction.” Later though the analysis of word2vec appears to center around categories with continous degrees of activity. With this section stating "To better understand the discovered concepts we will, ironically, appeal to a continuous projection.” Before visualising and discussing the embeddings in continous space. This does clearly undercut the narrative of the work which asserts binary categorisation to be a useful analytic tool.

Additionally the section on word2vec makes little discussion of the underlying geometry of the embedding space, despite this being the major focus of the experiments on artificially generated data. I think greater narrative clarity would make the paper much stronger: i.e. making the line of argumentation consistent across the sections, and more clearly relating the artificial and word2vec experiments.

C) While the writing is generally excellent, in many places the style is a bit informal. While accessably written papers are great, there’s a difference between accessability and formality -- some sentences could use tightening.

"In this form the CKA is not nice to optimize, but since our constraining set is a cone it is equivalent to the Frobenius distance between the double-centered matrices.” - What does it mean for something to be ’not nice’ to optimize? How do you know your constraining set is a cone?

"Parsimony is desirable, but there are several ways to define it.” - It would be good to start with introducing parsimony, I assume you mean ’simplicity’ akin to occam’s razor but it would be good to make this explicit.

"Since an exact fit is not always possible (or even desirable), we must define goodness of approximation.” - Why isn’t an exact fit desirable?

D) Novelty and relation to previous work.

It would be good to clarify that extensive previous work has tried to understand continuous representations in terms of symbols. It seems that the overall approach - approximating embeddings with category vectors - has been tried in previous work quite a bit, and that the novelty here is introducing more scalable methods that can work on higher-rank data. If this is the main contribution it would be great to make that clear at the end of the introduction, although if that is the main contribution that also raises some concern that the methods struggle to scale to 1000 samples. Additionally there is a clear relation between this and tensor product representations which might be worth mentioning.

E) Discussion Narrative.

This is a minor issue, but the discussion opts to introduce the notion of biological plausibility for the first time ("Both algorithms can in principle be implemented by a population of neurons with Hebbian input and recurrent synapses, and though we treat this as a curiosity for now, it raises the question of biological relevance.”). It might be best to either introduce this in the introduction and make it a part of the paper overall, or make the discussion relevant to the current content of the introduction.

**Questions:**

a) Could you clarify the geometric interpretation of the word2vec embeddings? What is the analogram here, and how does that aid interpretation? While I follow the intuition of aligning kernels between spaces, I struggle to see the broader impact of your geometric analysis of that process.

b) What is the benefit of interpreting embeddings in terms of discrete categories, if interpretation is ultimately a qualitative comparison of the continuous weights associated with those categories? I'm slightly unclear on the utility of the approach here. Is it intended to be useful for interpreting word embeddings (which appears to be a substantial claim of the analysis), if so relevance to other approaches to interpretability would be welcome - and some mention of the computation complexity of running the analysis on a sample of word2vec.

---

> ### Author Response · Authors · 2024-12-02
> **Response to reviewer wFCT**
>
> We thank the reviewer for the thorough evaluation of our manuscript. In addition to the general changes and comments, those changes addressing the reviewer's concerns are in orange.
>
> (A) Regarding loose and sometimes cumbersome notation: We have substantially refined our writing to introduce fewer symbols and to write equations in terms that are more likely familiar to the average reader. In particular, we have added sentences in Section 2.1 which provide an intuitive introduction to the CKA before the definition. Equation 4 has also been replaced by the those of Hopfield energy function, and while it is still rather dense owing to space constraints, but we feel it is important to include the equations in the main text for reproducibility.
>
> (B) Regarding many perspectives: This is an important narrative critique, and we have addressed it in part by substantially rewriting section 4. We (1) include a more explicit discussion of the geometry of the word2vec representations and how it relates to the discovered concepts through sparsity (Fig. 4b) and clarifying where the word2vec falls in the spectrum of low-rank to sparse concept spectrum of the synthetic tasks; (2) clarify the purpose of the plot in Fig. 4h; and (3) include a new analysis which is more clearly categorical in nature.
>
> (C) Regarding informal language: We have corrected instances of imprecise language, including those explicitly mentioned by the reviewer.
>
> (D) Novelty and contribution: Our rewritten prior work section, and new contributions section of the introduction now include more explicit connections to previous work, including the relevant work on tensor product representations of which we were not previously aware, and we highlight the ways that ours is conceptually and practically different. The practical contributions are bolstered by our much improved refinement algorithm.
>
> (E) Discussion: We have replaced the introduction of this concept with a more general summary of our results and the novelty of our approach to this problem.
>
> Questions:
>
> Q1: The analogram provides a way of visualizing the discovered concepts, essentially saying ``these four words are arranged as a square with some word-specific deviations''. By construction, the distances on the graph match the Hamming distance in the binary embedding, so an analogram can be used to quantitatively inspect the distances. The binary embedding can also be recovered from the graph by cutting at all edges with the same color, which will result in a partition of the nodes. The graph is unfortunately not always as useful as a dendrogram in hierarchical clustering, because it often cannot be plotted without lots of intersections.
>
> Q2: We hope that the new, more categorical analysis of the word2vec representations helps to clarify the utility of the categorical representations. While it is somewhat beyond the scope of this methods paper, we also expect that this technique could be useful as an unsupervised version of the ``concept bottleneck model'' which we now mention in the introduction -- briefly, the idea there is that a model's output is constrained to be computed as a function of some specified concepts, so that the model is ultimately more interpretable.
>
> Regarding the complexity, it took around 2 hours on a laptop to fit 1000 concepts on the 3841 words, using a relatively slow annealing schedule. It can take half as long with a faster annealing schedule (and a worse fit).

---

### Official Review · Reviewer_nLwJ · 2024-11-09

**Soundness:** 3
**Presentation:** 3
**Contribution:** 3
**Rating:** 5
**Confidence:** 3

**Summary:**

The paper explores identifying geometry representations to symbolically represent neural models and understand their reasoning patterns. The paper shows that the discovery of latent categories can be presented in a binary matrix factorization set-up, preserving mutual distances. Performance of the proposed framework is tested using low and high-rank representation matrices, showing potential to handle a variety of inputs.

**Strengths:**

- Detailed theoretical framework supported by experiments on simulated examples and neural word embeddings
- Discussion of performance of proposed method on low-rank and high-rank representations

**Weaknesses:**

- Limited discussion of key contributions and problem formulation in the introduction – while this is discussed across other sections in the paper, work would benefit from condensed clear problem formulation, main contributions and applications of the proposed framework
- Notation appears challenging to follow – for example p is used both for probabilities and number of points

**Questions:**

1. Have you tested your method on embeddings beyond word2vec? It would be good to see how the method generalizes on embeddings for data such as graphs or images
2. In section 2.1, you mention that the problem would be NP-hard to solve where uniqueness does not necessarily hold, could you clarify if this challenge was mitigated with use of CKA objective and proposed regularization?
3. In section 4, it is signaled that the proposed framework has potential to achieve CKA higher than 0.93, have you run experiments which support this?

---

> ### Author Response · Authors · 2024-12-02
> **Response to reviewer nLwJ**
>
> We thank the reviewer for the comments on our manuscript.
>
> (1) Regarding contributions: We agree that the paper suffered from lack of explicit discussion of the contributions, and so we have added a section about this to the introduction.
>
> (2) Regarding notation: Owing to our new algorithm formulation, the conflicting use of some letters has been resolved (although we still use $p_0$ to refer to a probability distribution, which we hope is clear from context), and we have endeavoured to clean up our notation in other parts of the paper by limiting the introduction of new symbols.
>
> Q1: While we do not have the space to include it in this paper, we have applied our method to neuroscience data (electrophysiology recordings) and to co-occurrence matrices from competitive pokemon games. A future version of this paper could highlight those analyses at the expense of some of the technical and theoretical parts.
>
> Q2: In the general case, unfortuantely we cannot expect to get around the NP hardness of the problem, in the sense of having an efficient algorithm with guarantees on solution quality. Of course this does not preclude good heuristics, as we have shown ours to be one. However, the regularization can help to mitigate the non-identifiability, by selecting for (in our case) the sparsest solution among the many equally good ones.
>
> Q3: The reviewer is correct to point out this statement, which was based on our experience with the solution quality achievable in smaller datasets but not on this one. We have indeed done better with our new algorithm (by about 0.01), but it is still an imprecise statement which we have removed.

---

> > ### Comment · Reviewer_nLwJ · 2024-12-03
> >
> > Thank you very much for providing further information on my queries. It would be great to see neuroscience and pokemon games experiments included in the future version of the paper, as this would complement to evidence of proposed method robustness. I am happy to increase my score to a 5, however I believe adding experiments beyond word2vec as part of appendix and further background on NP-hardness of the problem would make the work stronger.

---

### Official Review · Reviewer_bnzs · 2024-11-12

**Soundness:** 4
**Presentation:** 4
**Contribution:** 3
**Rating:** 6
**Confidence:** 4

**Summary:**

This paper proposes novel methods for discovering interpretable categorical structure within continuous data representations. The key idea is to find a binary embedding that preserves the original data geometry while representing data points as combinations of latent concepts. Authors introduce two algorithms: the Binary Auto-Encoder (BAE) for low-rank data, leveraging an approach inspired by Hopfield, and an Iterative Refinement algorithm for full-rank data, where sparsity regularization encourages simpler conceptual structures. Authors then visualizes the learned binary embeddings using novel "analograms." Experiments on simulated data and word embeddings demonstrate the method's ability to uncover meaningful latent concepts and their relationships.

**Strengths:**

* This paper makes a timely contribution motivated by LLMs' mechanistic interpretability. I learned a lot by how authors frames the problem of symbolic knowledge/algorithm extraction from distributed representation as geometry-preserving binary matrix factorization problem.

* Building on top of existing matrix factorization approaches, this paper proposes two concrete algorithms: (i) Binary Auto-Encoder (BAE)  and (ii) Iterative Refinement algorithm.

* BAE offers a computationally efficient alternative to existing methods, particularly for higher-rank, but still relatively low-rank scenarios. It notably avoids the restrictive Schur independence assumption required by some prior work.

* The Iterative Refinement algorithm provides a solution for the challenging full-rank scenario, which is less explored in prior work on binary matrix factorization.

* It was a smooth read. Presentation of the conceptual and algorithmic contributions are solid, where the "analograms" visualization provides a valuable tool for visualizing the learned binary embeddings.

* Claims are carefully evaluated on both simulated data and real-world word embeddings.

**Weaknesses:**

Below are some weaknesses. Overall, I appreciated how author themselves have honestly clarified many of the limitations of the proposed approaches in the draft.

* The theoretical justification for sparsity regularization feels a bit weak, given the inherent non-uniqueness and NP-hardness of the problem. Stronger guarantees on solution quality would be valuable.

* The Iterative Refinement algorithm, while clever, seems to struggle with larger datasets. That super-polynomial scaling is a real bottleneck, and we're not sure how practical this approach would be in many real-world scenarios.

* Interpreting the discovered concepts feels somewhat subjective and requires manual effort. A more automated approach to interpretation would be beneficial. (This is a broader issue for interpretability approaches though.)

* A more comprehensive comparison to existing methods would further strengthen the empirical results.

**Questions:**

* The scalability of Iterative Refinement seems like a real bottleneck. Have you considered any approximations or optimizations to improve its performance on larger datasets? What's the largest dataset you realistically think this algorithm could handle?

* I'd love to see a more thorough comparison with existing methods. Could you add more discussions regarding how your approach compares again recent works, e.g., The Geometry of Categorical and Hierarchical Concepts in Large Language Models by Kiho Park, Yo Joong Choe, Yibo Jiang, Victor Veitch?

* The connection between sparsity and analogram complexity isn't super clear. Have you investigated alternative regularization strategies that more directly target the complexity of the learned graph structure? I wonder if there are cases where sparsity might actually lead to more complex graphs.

---

> ### Author Response · Authors · 2024-12-02
> **Response to reviewer bnzs**
>
> We thank the reviewer for the thorough evaluation. In addition to the general changes in blue, changes which directly address these points are written in red.
>
> (1) Regarding the justification of the sparsity regularization: We agree that a more quantitative argument would be more compelling, unfortunately it was difficult to arrive at a useful result. For now, we have given a general intuitive argument about the size of the sparse solution space being generally smaller than the dense solutions (i.e. there are n-choose-k concepts with k nonzero elements). We have also included a more broad appeal to sparsity as a useful inductive bias, based on compressed sensing and some results in cognitive science.
>
> (2) Regarding the scalability: We hope that the improved refinement algorithm can address some of the concerns about scalablity, as it now scales competitively with two-layer autoencoders (with better results).
>
> (3) Regarding subjectivity: We agree with this general criticism of interpretability methods, and that ours, like those, can require a secondary method for interpretation. We note that the output of our algorithm can be used in the same way as those of a sparse autoencoder, e.g. asking a language model to interpret (we now mention this is section 4). We also offer our complementary analysis methods in section 4 as new interpretability tools for binary embbeddings in particular.
>
> (4) Regarding comparison: This was indeed an important analysis missing from our previous version, and we have addressed it by adding two baselines.
>
> Q1: Having addressed the main scalability bottleneck, we nevertheless still have trouble on very large datasets (>10000 or so). Three ways we can see to improve this are (1) better code, it is written in python which brings a big penalty from the for loops involved; (2) batching, i.e. computing the updates with a subset of the data; and (3) pre-clustering, i.e. reducinig the dataset size with a disjoint clustering algorithm.
>
> Q2: We thank the reviewer for pointing us to this very relevant paper, as it has encouraged us to discuss that literature more. We draw a connection in the second paragraph of the prior work section.
>
> Q3: It is good to point out that sparsity is not always tied to complexity. They are related, because the ``hidden'' nodes of the graph come from the intersection of each item's categories, e.g. if item X has labels {a,b} and item Y has {b,c}, then there will be a hidden node with labels {b}. If items have sparser labels they will have fewer intersections on average, but there are exceptions. We have also tried directly penalising label intersections, and found it more generally effective, but it has a more complicated implementation which we did not have the time to fully put into practice.

---

### Author Response · Authors · 2024-11-28
**Response to all reviewers**

We thank the reviewers for their careful reading of our manuscript. We would also like to apologise for the time taken in producing our revisions and writing this response; we were delayed due to personal circumstances of one of the authors, and we are grateful for the extended discussion period during which we can respond properly to each reviewer.

There are some common problems that were identified by multiple reviewers, and we will address them here. **Issues raised by individual reviewers will be addressed in replies in the coming days.**

1) Practicality of Algorithm 2

This is where we can offer the most improvement. In the time since submission, we have found that the binary iterative refinement algorithm, which we mention in the main text and had included in the Appendix, is much more robust and effective than anticipated. Furthermore, it is easier to explain, and can be integrated naturally with Algorithm 1. In short, we still optimize one row of **S** at a time, but we do so directly instead of using a continuous relaxation. While we sacrifice the convex updates of the continuous algorithm, this provides more robustness to noise, since it fixes the maximum rank of the factorization. We solve the resulting quadratic binary optimization using Hopfield networks, which also allows us to take advantage of the sparsity of **S** (since it is just iterated matrix-vector multiplication). Section 3.2 has been updated to explain the new algorithm, Figures 3 and 4 have been updated to reflect the new algorithm, and a section has been added to the Appendix providing derivations of the weight and bias terms of the Hopfield network. The result is that **the improved algorithm scales very well across our simulated range of dataset parameters**.

2) Novelty/utility of work

We address this question with a substantially reworked prior work section, a new contributions section, and a comparison against other methods.

To summarize our changes to the prior work: We replaced the admittedly technical connections to PCA and overlapping K-Means with more relevant conceptual connections to neuro-symbolic approaches, mechanistic interpretability, and distance-preserving hashing. We also provide a practical connection to a popular class of models known as "concept bottleneck models" which we recently discovered.

To summarize our contribution section: Our contributions are both conceptual and practical. (a) Conceptually, while the problem of (semi) binary matrix factorization has been studied before in applied math, to our knowledge it has not been used to formalize the search for general discrete structure in continuous representations. The simplicity and generality of the approach is itself a conceptual contribution. (b) Practically, our algorithm for semi-BMF is able to combine benefits of previous approaches, both robustness and speed. We also offer new visual tools (analogram).

We also compare against two gradient-based approaches: a tanh autoencoder and a Bernoulli variational autoencoder. The main difference between the two is the presence of a binary entropy term in the Bern-VAE coming from the KL divergence. We make the number of parameters and iterations comparable across models, taking the best hyperparameters (learning rate, annealing schedule, etc.) for each model. We chose not to use extant SBMF algorithms, for technical reasons (they are tricky to implement, and most do not scale very well), because they have already been benchmarked recently, and because they do not apply to most data we consider.

3) Relevance of word embedding application

It is a fair point that the analysis of word2vec is under-explored and does not adequately demonstrate potential uses of the binary representation. Thus we have substantially reworked the narrative of this section, and provided a new analysis

Regarding narrative, we now highlight that the continuous projection of the concept embeddings is primarily as a tool for interpreting the discovered concepts. As a visualization itself, we also emphasise its added interpretability owing to being interval valued with known extreme points.

We have added additional analysis of the word2vec concepts, in which we try to give a sense of *global* structure, and is more categorical in nature. Briefly, we searched for pairs of dataset-level concepts which have high mutual parallelism (illustrated in the new Figure 4) of their centroids. Two concepts define four clusters (one for each combination of values), and the data are well-separated in the discovered feature dimensions. Given the concepts, we then look for quadruplets of words which have some analogy structure to them.

Finally, we note that these are only visualization tools -- there are also practical uses for binary representations, such as hashing and the aforementioned ``concept bottleneck models'', which we do not showcase in this paper, but we do now mention them in this section.

---

### Meta-Review · Area_Chair_z69K · 2024-12-21

**Metareview:**

This paper studies the very important and challenging task of inferring discrete structure represented within continuous representations.

As it stands, even though the Authors' rebuttal resulted in some scores being raised, it appears the majority of the reviewers are still leaning on the rejection side. From my read of the reviews and the paper, it appears that one key point of improvement is that more experimental work could strengthen the paper's contributions (as already pinpointed e.g. by Reviewer nLwJ).

All taken into account, I recommend this paper for rejection. The Reviewers did not oppose this decision.

**Additional Comments On Reviewer Discussion:**

It is unfortunate that the Authors had personal issues limiting the amount of work they could do in response. I regret to hear of the existence of these circumstances and wish the Authors all the best going forward.

That being said, as per my comments above, it is my belief that it would not do the paper justice to accept it in its present form---a thorough round of revisions is recommended. I reiterate that the topic of the work is highly valuable, and as such I am hopeful the Authors will continue to pursue it!

---

### Decision · Program_Chairs · 2025-01-22

Reject